# Mummified baboons reveal the far reach of early Egyptian mariners

**Nathaniel J Dominy[1]\*, Salima Ikram[2], Gillian L Moritz[1†], Patrick V Wheatley[3], John N Christensen[3], Jonathan W Chipman[4], Paul L Koch[5]**

[1]Departments of Anthropology and Biological Sciences, Dartmouth College, Hanover, United States; [2]Department of Sociology, Egyptology, and Anthropology, American University in Cairo, New Cairo, Egypt; [3]Center for Isotope Geochemistry, Lawrence Berkeley National Laboratory, Berkeley, United States; [4]Department of Geography, Dartmouth College, Hanover, United States; [5]Department of Earth and Planetary Sciences, University of California, Santa Cruz, Santa Cruz, United States

**\*For correspondence:**
nathaniel.j.dominy@dartmouth.edu

**Present address:** [†]Department of Physiology and Cell Biology, University of Nevada, Reno, United States

**Competing interests:** The authors declare that no competing interests exist.

**Abstract** The Red Sea was witness to important events during human history, including the first long steps in a trade network (the spice route) that would drive maritime technology and shape geopolitical fortunes for thousands of years. Punt was a pivotal early node in the rise of this enterprise, serving as an important emporium for luxury goods, including sacred baboons (*Papio hamadryas*), but its location is disputed. Here, we use geospatial variation in the oxygen and strontium isotope ratios of 155 baboons from 77 locations to estimate the geoprovenance of mummified baboons recovered from ancient Egyptian temples and tombs. Five Ptolemaic specimens of *P. anubis* (404–40 BC) showed evidence of long-term residency in Egypt prior to mummification, consistent with a captive breeding program. Two New Kingdom specimens of *P. hamadryas* were sourced to a region that encompasses much of present-day Ethiopia, Eritrea, and Djibouti, and portions of Somalia and Yemen. This result is a testament to the tremendous reach of Egyptian seafaring during the 2nd millennium BC. It also corroborates the balance of scholarly conjecture on the location of Punt.

## Introduction

The sacred baboon (*Papio hamadryas*) was a recurring motif in ancient Egyptian art and religion, from Predynastic statuettes to later mortuary traditions, including wall paintings, reliefs, amulets, and statues—a tradition exceeding 3000 years (*Figure 1*). In most cases, *P. hamadryas* was the embodiment of Thoth, a deity associated with the moon and wisdom. It is a rare example of apotheosis among nonhuman primates. The archetype of this manifestation—a male baboon in a seated posture, hands on knees, and often surmounted with a lunar disc or crescent—was strikingly consistent for millennia. Figurines of seated baboons even spread into the Levant and across the Mediterranean during the Middle Bronze Age, but the biological realism diminished with increasing distance from Egypt, becoming symbolic renderings rather than species-specific representations (*Dothan and Regev, 2011*). Such a pattern invites two complementary interpretations: first, that ancient Egyptian artists were concerned with species-level realism and second, that they were themselves witness to the plants and animals in their works. Accordingly, some ecologists have viewed the artistic record of Egypt as a biological survey and used it to assess ecosystem stability through time (*Yeakel et al., 2014*).

Yet, the Holocene fossil record of Egypt is devoid of any monkey species, let alone *P. hamadryas* (*Geraads, 1987*). Gaps in the fossil record are common, of course, and seldom conclusive on the question of regional absence, but the premise is reinforced by ecological modeling (*Chala et al., 2019*), which indicates little appreciable change to the distributions of baboons during the past

**eLife digest** Strontium is a chemical element that can act as a geographic fingerprint: its composition differs between locations, and as it enters the food chain, it can help to retrace the life history of extant or past animals. In particular, strontium in teeth – which stop to develop early – can reveal where an individual was born; strontium in bone and hair, on the other hand, can show where it lived just before death. Together, these analyses may hold the key to archaeological mysteries, such as the location of a long-lost kingdom revered by ancient Egyptians.

For hundreds of years, the Land of Punt was one of Egypt's strongest trading partners, and a place from which to import premium incense and prized monkeys. Travellers could reach Punt by venturing south and east of Egypt, suggesting that the kingdom occupied the southern Red Sea region. Yet its exact location is still highly debated.

To investigate, Dominy et al. examined the mummies of baboons present in ancient Egyptian tombs, and compared the strontium compositions of the bones, hair and teeth of these remains with the ones found in baboons living in various regions across Africa. This shed a light on the origins of the ancient baboons: while some were probably raised in captivity in Egypt, others were born in modern Ethiopia, Eritrea, Djibouti, Somalia and Yemen – areas already highlighted as potential locations for the Land of Punt.

The work by Dominy et al. helps to better understand the ancient trade routes that shaped geopolitical fortunes for millennia. It also highlights the need for further archaeological research in Eritrea and Somalia, two areas which are currently understudied.

20,000 years. Such evidence distinguishes *P. hamadryas* as the only animal member of the Egyptian pantheon that is naturally absent from Egypt today and during antiquity. Setting aside the puzzling question of why ancient Egyptians deified *P. hamadryas* (*Thomas, 1979*), the level of reverence was sufficient to justify the importation, husbandry, and mummification of it and another species, *P. anubis*, the olive baboon. (See *Box 1* on the biogeography of *Papio*).

The distributions of *P. anubis* and *P. hamadryas* differ, a fact that bears on the trade networks that supplied living baboons to Egypt (*Figure 2*). Assuming little change to these distributions over the past 5000 years (*Chala et al., 2019*), then *P. anubis* was readily available via overland trade, whereas *P. hamadryas* was more practically obtained via maritime trade—indeed, the distinction is apparent on temple walls, insofar as *P. hamadryas* is the only baboon associated with long-distance seafaring (*Figure 1c*). The problem that motivates us lies with the contested nautical range of Egyptian ships (*Wicker, 1998*), and the untapped potential of baboons to reveal the geography of early maritime trade in relation to Punt, a fabled emporium. Our goal is to use isotopic mapping to determine the geoprovenance of mummified baboons recovered from New Kingdom temples and Ptolemaic catacombs; but first, a brief chronology is necessary to introduce the specimens, as the cultural context and availability of each species varied through time (*Figure 2*).

## Predynastic specimens

Skeletal remains of *P. anubis* (n = 16) are present at the cemetery site HK6, Hierakonpolis (3750–3550 BC) (*Van Neer et al., 2017*). The assemblage contains adults and juveniles of both sexes, buried singly and in groups. A juvenile baboon (age: 4–5 years) was interred with an adolescent person (age: 10–15 years), suggesting status as a pet (*Van Neer et al., 2004*). The prevalence of skeletal pathologies points to physical abuse under captive conditions (*Van Neer et al., 2017*). The absence of *P. anubis* at every other Egyptian site of this time period, both archaeological and non-archaeological, together with the presence of two elephants at HK6, is interpreted as evidence of overland animal trade with peoples farther south in present-day Sudan (*Van Neer et al., 2004*).

## New Kingdom specimens

In 1837, the British Museum purchased two mummified baboons from the estate of Henry Salt, British Consul-General in Egypt from 1816 to 1827. The given provenance of each specimen—Temple of Khons (EA6736; *Figure 3a*) and Thebes (EA6738; *Figure 3c*)—is itself sufficient to impute a (late) New Kingdom origin, an inference bolstered by two lines of typological evidence illustrated in

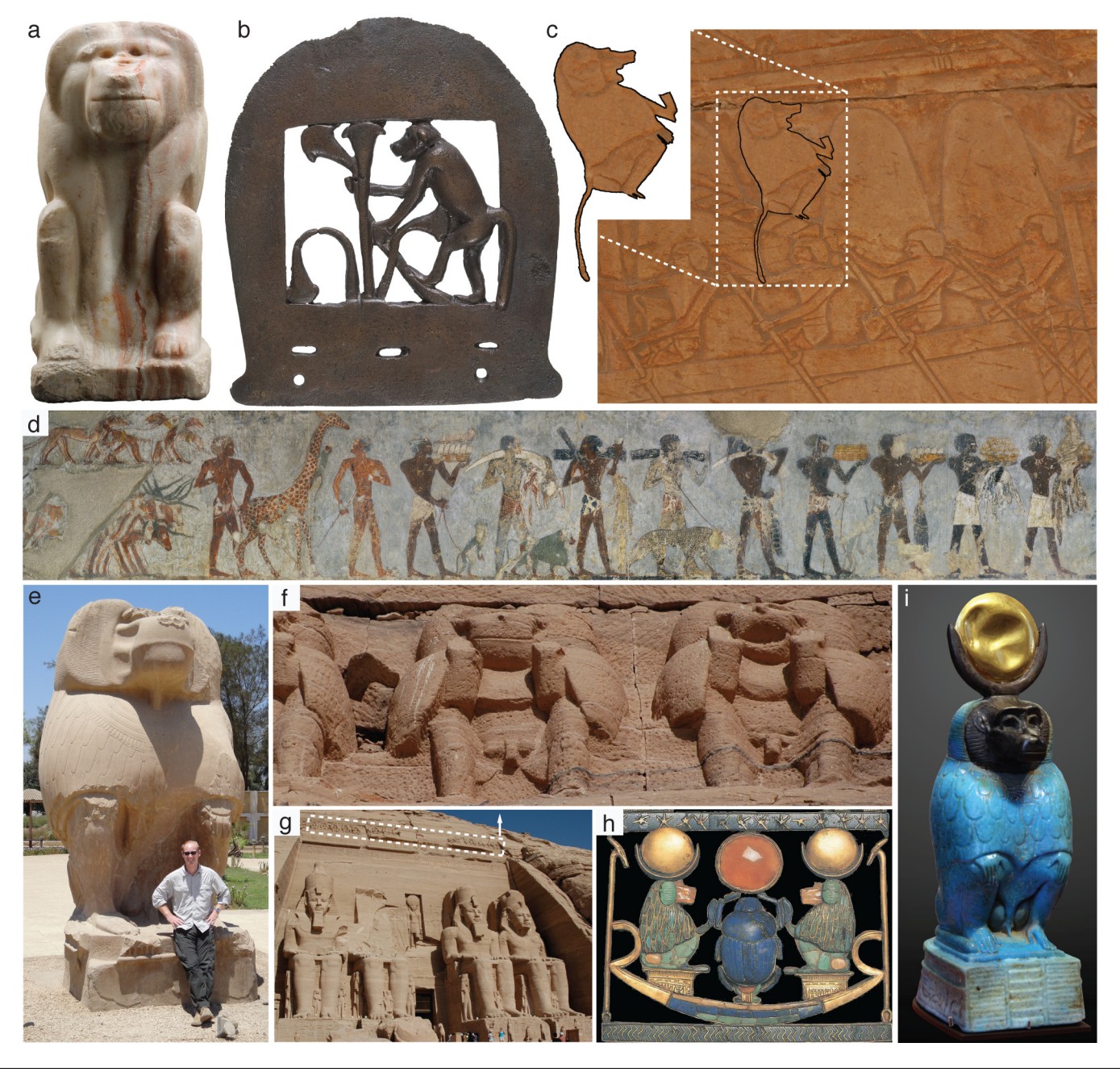

**Figure 1.** Egyptian iconography of *Papio hamadryas*, a tradition exceeding 3000 years. (**a**) Statuette inscribed with the name of King Narmer, Early Dynastic Period, 1st Dynasty, ca. 3150–3100 BC (no. ÄM 22607, reproduced with permission from the Ägyptisches Museum und Papyrussammlung, 2020, under the terms of a CC0 1.0 license. (**b**) Bronze axe head, Middle Kingdom, 12th or 13th Dynasty, ca. 1981–1640 BC (no. 30.8.111, reproduced with permission from The Metropolitan Museum of Art, 2020, under the terms of a CC0 1.0 license. (**c**) Reliefs at the mortuary temple of Queen Hatshepsut [Deir el-Bahari]. A hamadryas baboon sits in the rigging of a ship. It is one of five being imported from Punt; New Kingdom, 18th Dynasty, ca. 1473–1458 BC. (**d**) Wall painting in the mortuary chapel of Rekhmire (TT 100), Vizier to Tuthmose III and Amenhotep II. A baboon (*P. hamadryas*) is shown as tribute in a procession from Nubia. Three vervets (*Chlorocebus aethiops*) are also illustrated, one of which climbs the neck of a beautifully rendered giraffe; New Kingdom, 18th Dynasty, ca. 1479–1425 BC. (**e**) Large 35-ton statue at Hermopolis Magna (author NJD shown for scale); erected by Amenhotep III, New Kingdom, 18th Dynasty, ca. 1370 BC. (**f**) Frieze of baboons on the east-facing facade of the rock-cut temple of Abu Simbel (**g**), New Kingdom, 18th Dynasty, ca. 1265 BC. The raised arms are interpreted as a posture of adoration toward the rising sun, whereas the open mouth may represent vocal behavior (**te Velde, 1988**). (**h**) Pectoral necklace of Tutankhamun; baboons are surmounted with lunar disks and simultaneously adoring the central solar disk, a rare combination of two stereotypical postures; New Kingdom, 18th Dynasty, ca. 1341–1323 BC (no. JE 61885, Museum of Egyptian Antiquities). (**i**) Faience figurine and exemplary representation of Thoth: a male *P. hamadryas* in a seated posture, hands on knees, and

*Figure 1 continued on next page*

surmounted by a lunar disc, Ptolemaic period, 332–30 BC (no. E 17496, Musée du Louvre).
© 2020, Rama, Creative Commons Attribution-ShareAlike 3.0. *Figure 1I* is reproduced with permission from Rama, 2020, under the terms of the Creative Commons Attribution-ShareAlike 3.0; this image is not distributed under the terms of the CC0 1.0 license, and further reproduction of this image panel should adhere to the terms of the CC-BY-SA 3.0 license.
© 2019, Sandro Vannini. All rights reserved. *Figure 1D* is reproduced with permission from Sandro Vannini, 2019; this image is not distributed under the terms of the CC0 1.0 license, and further reproduction of this image panel would need permission from the copyright holder.

*Figure 2*. First, the species attribution is *P. hamadryas* (*Anderson and de Winton, 1902*), the subject of numerous New Kingdom wall paintings and reliefs, some of which depict the act of importation from distant polities (*Appendix 1—figure 1*). Second, the seated posture and right-curled tail of EA6736 resembles the state of five mummified *P. hamadryas* found in the Valley of the Kings in 1906 (*Davis et al., 1908*; *Lortet and Gaillard, 1909*). The baboon-bearing tombs (KV50, KV51, KV52) are attributed on the basis of proximity to Amenhotep II (ca. 1427–1401 BC) or King Horemheb (1319–1307 BC), both members of the 18th Dynasty (*Ikram, 2005*). A third line of evidence concerns ante-mortem canine extraction (*Figure 3*), an idiosyncrasy that unites EA6736 and EA6738 with other specimens from the New Kingdom period. (See *Box 2* and *Appendix 1—figure 2* on the New Kingdom practice of canine extraction).

## Late and Ptolemaic specimens

Excavations in 1968 revealed a catacomb in the sacred animal necropolis of North Saqqara (*Figure 2* and see *Appendix 1—figure 3*), containing the remains of olive baboons (*P. anubis*; n = 143), vervets (*Chlorocebus aethiops*; n = 2), and Barbary macaques (*Macaca sylvanus*; n = 21) (*Goudsmit and Brandon-Jones, 1999*). Inscriptions indicate use from ca. 404 to 40 BC (*Davies, 2006*). The sample of *P. anubis* includes every age class, a demographic pattern consistent with captive breeding; however, the 2:1 ratio of males to females indicates sex-biased internment (*Goudsmit and Brandon-Jones, 1999*). An outstanding feature of the site is the preservation of six obituaries written in demotic script. In one example, an animal named Harnufi (and later, Djeho-the-baboon, a sacred by-name) was imported into Egypt in year 6 of Ptolemy V (200/199 BC) and buried on 06 September 168 BC (*Ray, 2011*), a minimum captive span of 31 years. In other examples, we learn that animals were brought from Alexandria or the temple estate of Ptah-under-his-moringa-tree in Memphis, where they were born, housed, and mummified (*Ray, 2011*). Most of the baboons (82%) have craniodental or developmental anomalies associated with vitamin D deficiency and prolonged indoor confinement (*Goudsmit and Brandon-Jones, 1999*).

Mummified baboons are also known from the subterranean galleries of Tuna el-Gebel (*Figure 2* and see *Appendix 1—figure 3*), dating from the reigns of Ptolemy I and II (303–246 BC) (*von den Driesch, 1993b*). The assemblage was described in 1989 (82 specimens of *P. anubis*, *Kessler, 1989*) and again in 2005 (>200 specimens, including one *P. hamadryas* [*Kessler and Nur el-Din, 2005*]). The total number of burials is estimated at ~2000 (*Kessler and Nur el-Din, 2005*). The initial data set contained a comparable number of adult males (n = 34) and females (n = 28), which hints at captive breeding, yet it was devoid of animals < 2 years of age. Severe bone deformations and chronic degenerative pathologies in 38 individuals suggest extended captivity under abject conditions (*von den Driesch and Boessneck, 1985*; *von den Driesch, 1993a*; *Nerlich et al., 1993*).

This treatment of *P. anubis* stands in stark contrast with the veneration directed toward *P. hamadryas*, as attested by massive statues at Hermopolis Magna, a temple 6 km east of Tuna el-Gebel (*Figure 1e*). The reasons for this distinction—the mummification of *P. anubis* vs. the iconography of *P. hamadryas*, both in the service of Thoth—are uncertain, but it was widespread during this time period.

## Study design and aims

Isotopic mapping (isomapping) is used to project geospatial variation in the isotope ratios of a given element, together with isotope ratios in biological tissues such as hair, bones, and teeth. The method is useful for estimating the geographic origin and subsequent movement of mobile organisms. Here, our goal is to use oxygen and strontium isomapping to estimate the source of baboons

## Box 1. Biogeography of baboons.

The taxonomy of baboons is a topic of enduring debate, with major types classified as either species or allopatric subspecies of the superspecies *Papio hamadryas* (**Jolly, 1993**). The distinction is essentially a matter of philosophy, and here we follow *Zinner et al., 2013* by recognizing six phenotypically distinct allotaxa as species: the sacred or hamadryas baboon (*P. hamadryas*), the olive baboon (*P. anubis*), the yellow baboon (*P. cynocephalus*), the chacma baboon (*P. ursinus*), the Kinda baboon (*P. kindae*), and the Guinea baboon (*P. papio*). Still, these allomorphs interbreed freely in areas of sympatry, and molecular studies report widespread mitochondrial paraphyly within and between the northern and southern clades, suggesting a long history of introgressive hybridization.

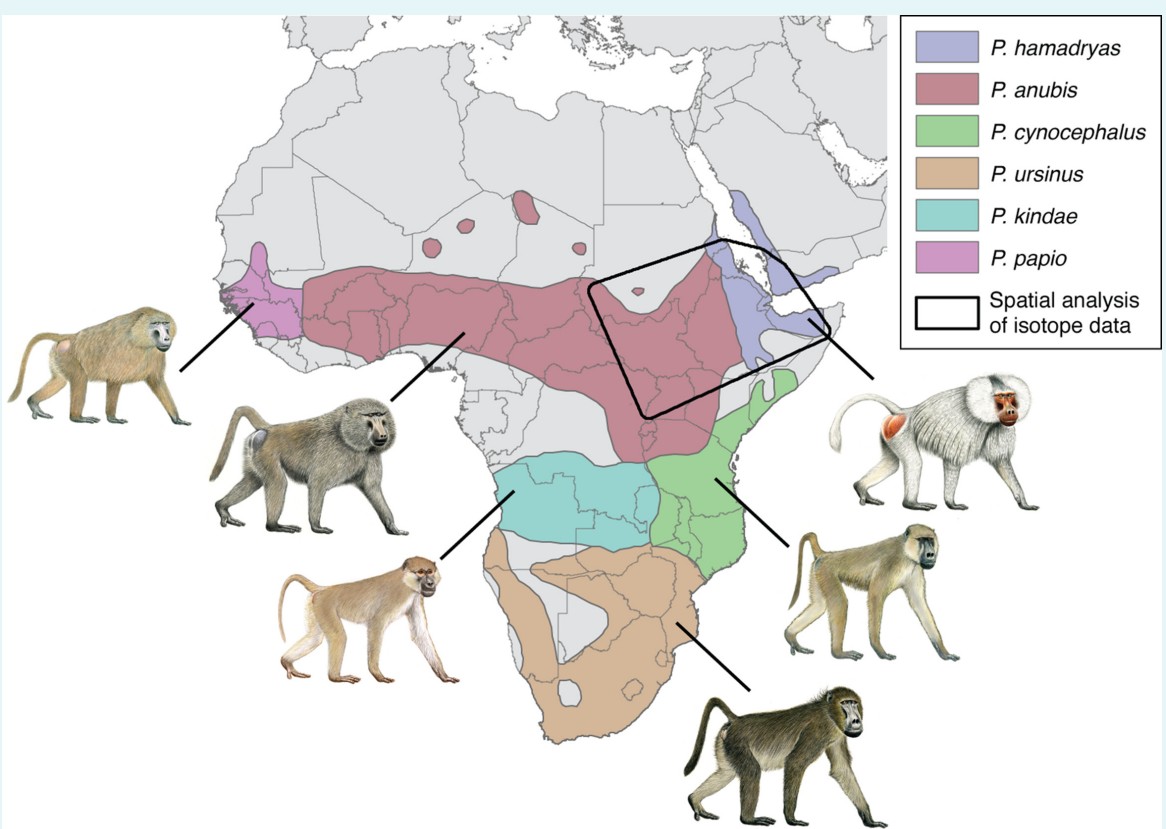

**Box 1—figure 1.** Modern geographic distribution of baboons (*Papio*) in Africa and southwest Arabia.
The polygon illustrates our area of geospatial analysis, which encompasses regions inhabited by sacred baboons (*P. hamadryas*) and olive baboons (*P. anubis*).

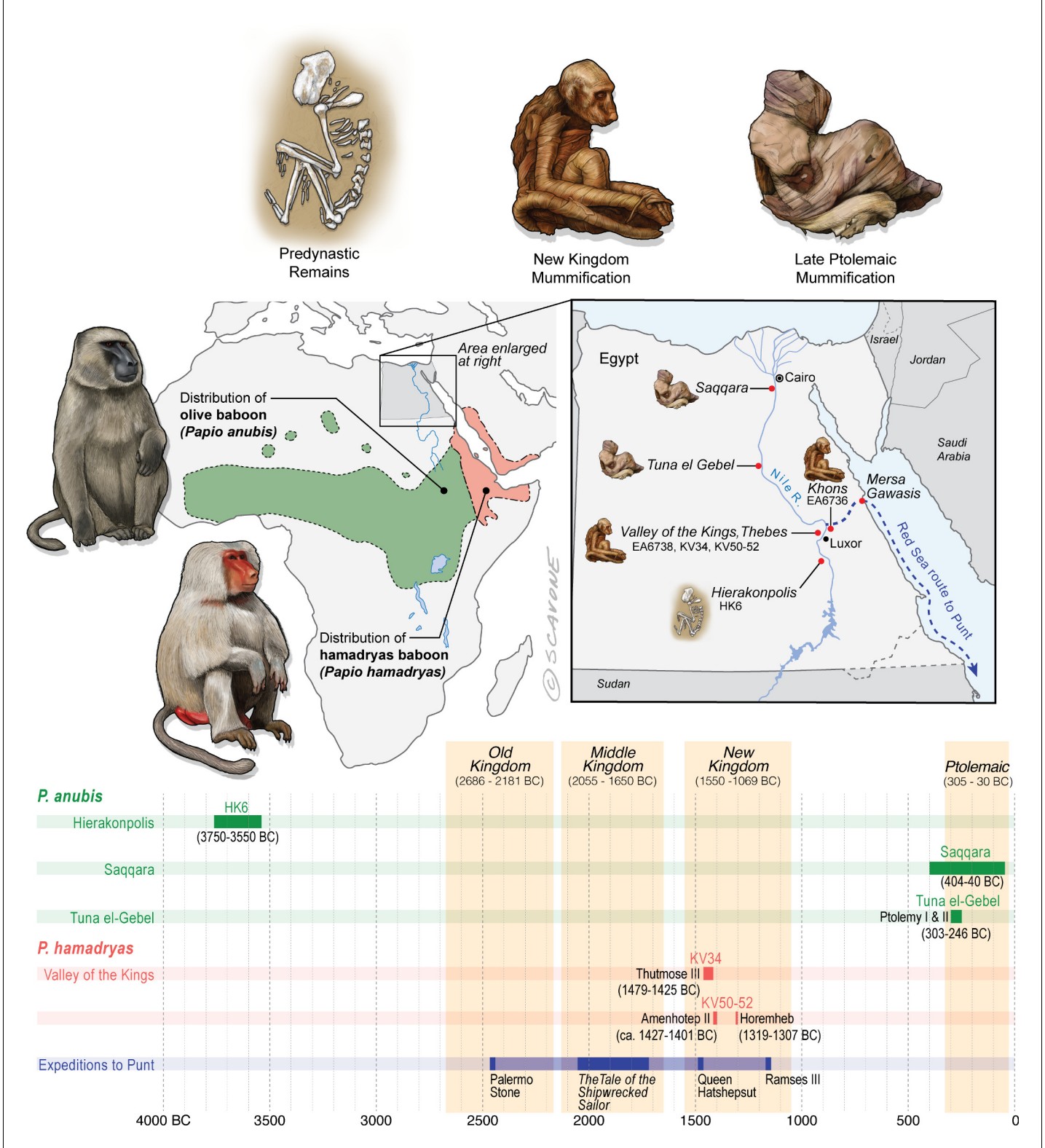

**Figure 2.** Egypt lies well beyond the distributions of *P. anubis* and *P. hamadryas*, and there is no evidence of natural populations in Egypt during antiquity. The remains of baboons in Egypt are therefore interpreted as evidence of foreign trade. This figure puts the present study specimens— EA6736, EA6738, and those of Saqqara—into context by illustrating spatiotemporal variation in the preservation of baboons, emphasizing differences in taxonomy, wrapping, and deposition (i.e., burials, tombs, or catacombs). Horizontal bars represent the temporal spans of baboon-bearing archaeological sites and known expeditions to Punt. Every New Kingdom specimen of *P. hamadryas* is penecontemporaneous with expeditions to Punt

*Figure 2 continued on next page*

*Figure 2 continued*

and associated with a royal temple or tomb. The quality of New Kingdom mummification is often extremely high, in part because the limb and tail elements were wrapped individually. Excluded from this figure is a baboon of uncertain taxonomy and disposition found buried in a palace at Tell el-Dab'a, and dating from the 18th Dynasty (*von den Driesch, 2006*). Mersa Gawasis is a Middle Kingdom harbor and port that was used to launch and receive seafaring voyages to Punt.

Illustration by William Scavone, Kestrel Studios.

in ancient Egypt. The physiology of baboons is particularly well suited to this approach, although the above evidence of prolonged captivity poses a practical challenge.

Baboons sweat under conditions of thermal stress, turning to sources of meteoric water (pools, streams, lakes) to replenish their body water and maintain homeothermy. Termed 'obligate drinking', this daily behavior produces a strong, mechanistic link between the oxygen isotope ratios ($^{18}$O/$^{16}$O) of precipitation and the baboons themselves (*Moritz et al., 2012*). Such a relationship is advantageous when using oxygen isotope values ($\delta^{18}$O) to trace the regional movements of animals (*Bowen et al., 2005b*), including primates (*Ehleringer et al., 2008*); however, it can be difficult to determine region-of-origin from $\delta^{18}$O values alone, a problem that is only exacerbated with increasing time depth.

Another problem is twofold: first, each mummified baboon in our sample died under captive conditions, and it follows that they were provisioned with Nile-sourced drinking water for some portion of their lives (*Touzeau et al., 2013*), which is expected to undermine the geospatial value of $\delta^{18}$O values in tissues that turn over quickly; and second, we have a limited sample of mixed tissues—hair from EA6736 (*Figure 3a*); hair, bone, and enamel from EA6738 (*Figure 3c*); and bone from five skulls, each from the Baboon Catacomb of North Saqqara—representing varying periods of temporal integration. Bone and enamel apatite incorporate years of dietary behavior at different life history stages, whereas hair keratin records information over weeks and months prior to death depending on length.

The enamel of EA6738 holds promise because the mineralization of its permanent incisors occurred early in life, between 1 and 3 years of age (*Dirks et al., 2002*); and because, in contrast to bone mineral, enamel apatite is resistant to postmortem alteration (*Hoppe et al., 2003*). It invites measurement of strontium isotope ratios—the $^{87}$Sr/$^{86}$Sr ratio of soils enters foodwebs through leaching by surface waters—together with those of modern baboons for the purpose of isomapping (*Bataille et al., 2020*). Accordingly, we collected 155 tissue samples from modern baboons representing 77 discrete locations (*Appendix 1—tables 1* and *2*), and employed a dual-isotope ($\delta^{18}$O; $^{87}$Sr/$^{86}$Sr) approach to estimate the geoprovenance of baboons imported into ancient Egypt.

## Results and discussion

### Late and Ptolemaic specimens

We measured strontium isotope ratios in the bone apatite of *P. anubis* (n = 5) recovered from the Baboon Catacomb of North Saqqara and accessioned in the Petrie Museum of Egyptian Archaeology, University College London (*Appendix 1—figure 3*). The $^{87}$Sr/$^{86}$Sr ratios were strikingly invariant ($\bar{x} = 0.70779 \pm 0.00007$; *Appendix 1—table 3*) and indistinguishable from those of ancient Egyptians and Theban carbonate rocks (*Figure 4*). This finding could result from Sr exchange (diagenesis) with the catacomb itself—indeed, the present samples were probably sourced from piles of macerated, jettisoned bones, the likely result of fourth-century Roman vandalism (*Appendix 1—figure 3*). Diagenesis from contact with the catacomb walls or dust is therefore plausible (*Hoppe et al., 2003*), but it fails to account for the distinctly non-local $^{87}$Sr/$^{86}$Sr ratio of a vervet in the same assemblage (*Appendix 1—tables 3*).

In general, the aridity of Egyptian tombs acts to preserve the in vivo (biogenic) Sr isotope values of bone apatite (*Touzeau et al., 2013*). If the present Sr values from *P. anubis* are biogenic then they indicate a uniform diet of provisioned food and water that was essentially identical to that of Egyptians living in the Nile Valley. The significance of this interpretation is twofold: first, it corroborates written and osteopathological evidence of prolonged captivity in Egypt, and, by extension, the possibility of a baboon breeding program in Memphis (*Ray, 2011*); and second, it thwarts our goal

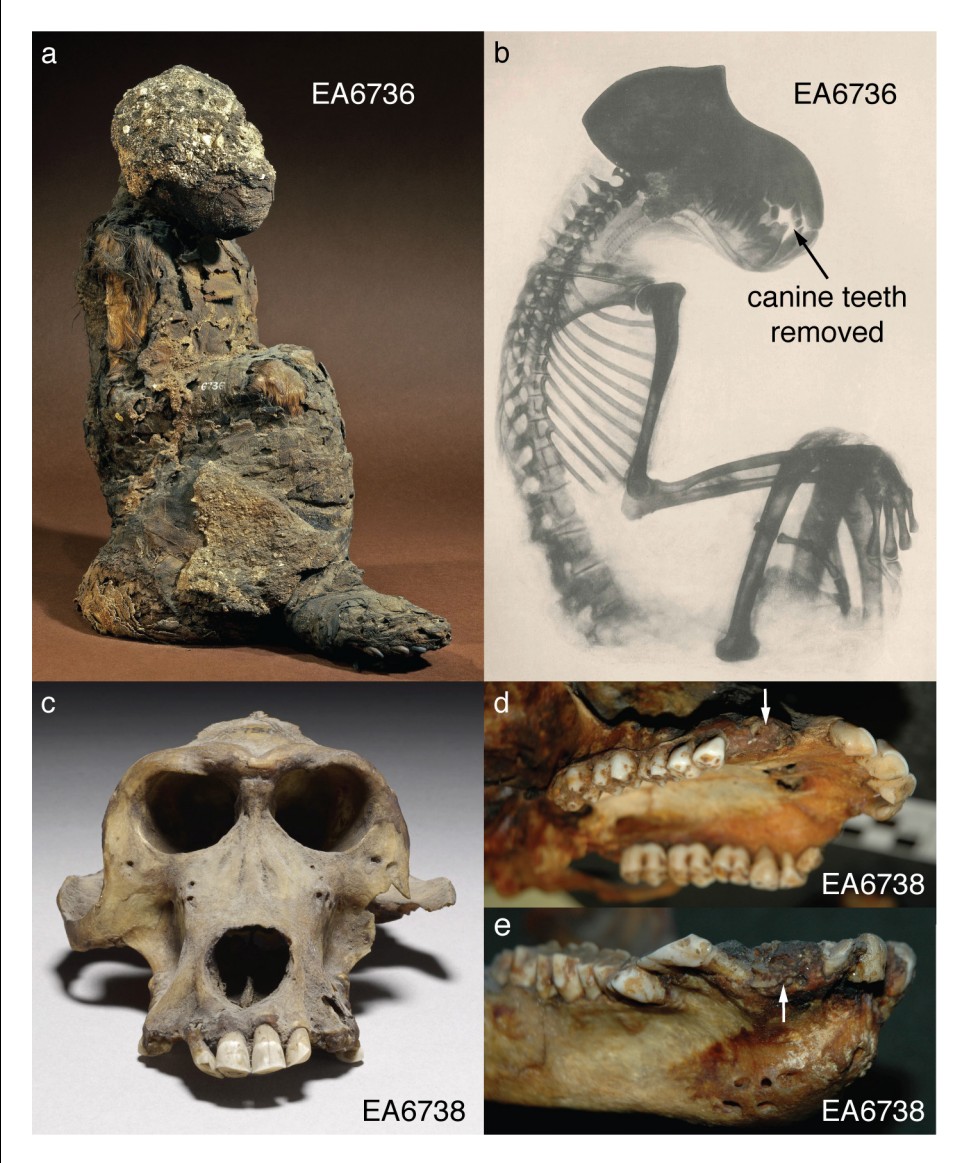

**Figure 3.** The British Museum holds two mummified baboons with New Kingdom attributions. (a) EA6736 is attributed to *P. hamadryas* (*Anderson and de Winton, 1902*). The present analysis is based on six strands of hair sourced from the upper right arm. (b) EA6736 was the subject of an early radiograph in 1899, which revealed the absence of four canine teeth (*Anderson and de Winton, 1902*). (c) EA6738 is also attributed to *P. hamadryas* (*Anderson and de Winton, 1902*), and it was the source of three tissue types: hair, bone, and enamel. It is also devoid of canines Ossification of the corresponding maxillary (d) and mandibular (e) alveolar sockets is evidence that the animal survived the procedure for many years.

## Box 2. Canine teeth and their extraction.

The extraction of canines in vivo was a prudent safety precaution—a single bite from an adult male baboon can cut human thigh muscle to the bone (**Walker, 1984**), and the available evidence points to regular interactions between humans and baboons at close quarters. Scores of New Kingdom paintings depict *P. hamadryas* working with people, often in a utilitarian role (as police animals; as fruit-harvesters [**d'Abbadie, 1964**; **d'Abbadie, 1965**; **d'Abbadie, 1966**; **Houlihan, 1997**; **Osborn and Osbornová, 1998**; **Deputte and Anderson, 2009**]), whereas some mummified individuals are interpreted as royal pets (**Ikram, 2004**) due to the high quality of mummification (**Lortet and Gaillard, 1909**) and close association with royal tombs (**Davis et al., 1908**). Canine extraction is evident in specimens of *P. hamadryas* from three royal tombs—KV34 (Thutmose III), KV51, and KV52—in the Valley of the Kings. It is also evident in the assemblage of baboons from nearby Gabanet el Giroud (**Lortet and Gaillard, 1907**), containing *P. hamadryas* (n = 4 females, two males), *P. anubis* (n = 5 females, one male), and five indeterminate juveniles. It has some New Kingdom affinities (**Lortet and Gaillard, 1907**), but radiocarbon dating of one specimen (MHNL 90001206; Musée des Confluences, Lyon) produced a date range of 803–544 BC (**Porcier et al., 2019**). The assemblage features numerous osteopathologies and examples of burial in lieu of formal mummification. In general, scholars tend to view canine extraction as a measure of human esteem because it speaks to close human-baboon interactions.

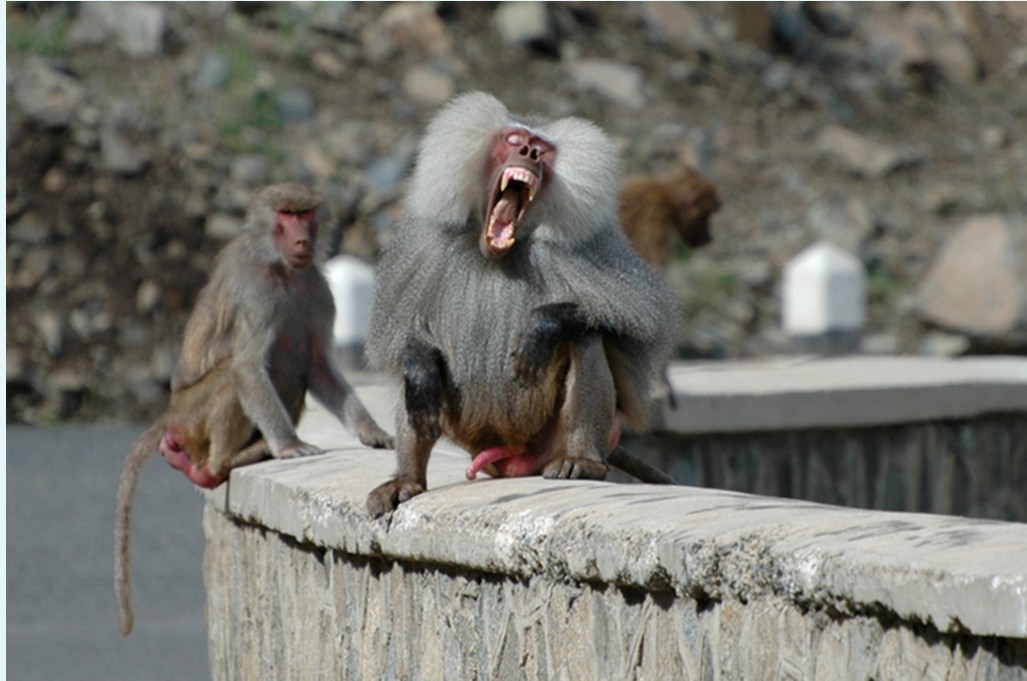

**Box 2—figure 1.** Adult males of *Papio hamadryas* have large, formidable canine teeth that can be used to telling effect.

Photograph by author NJD.

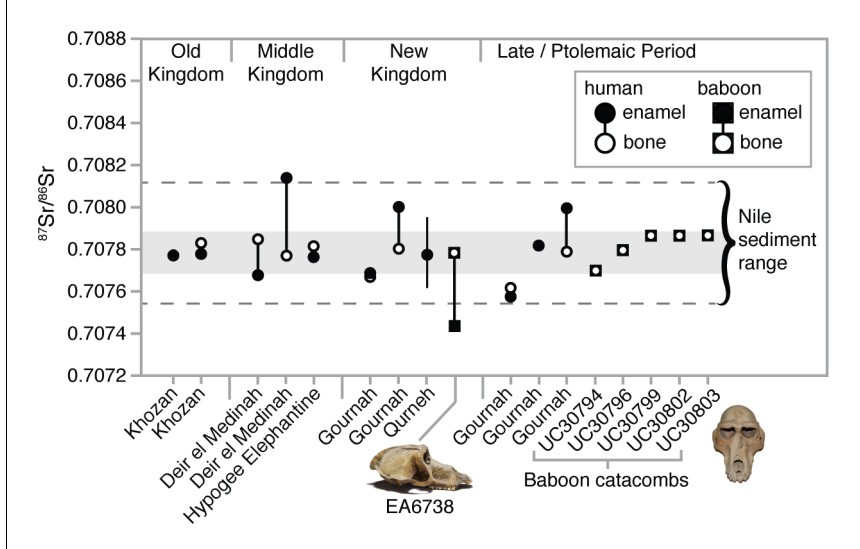

**Figure 4.** Strontium isotope ratios ($^{87}$Sr/$^{86}$Sr) of enamel and bone from humans (***Touzeau et al., 2013***) and baboons—EA6738 (*Papio hamadryas*; Thebes) and UC30794-UC30803 (*P. anubis*; Baboon Catacomb, North Saqqara)—recovered from Egyptian sites. The data set from Qurneh represents 15 people (mean ± 1 SD; source: ***Buzon and Simonetti, 2013***). The gray-shaded region represents the range of Theban carbonate rocks, whereas the dashed lines define the range of Nile sediments (source: ***Touzeau et al., 2013***). The divergent strontium isotope ratios of EA6738 are telling: the composition of enamel indicates early-life mineralization outside of Egypt, whereas the composition of bone indicates complete diagenesis or many years of living in Egypt prior to death and mummification.

of determining the geoprovenance of this sample. Fortunately, our analysis of two New Kingdom specimens proved more revealing.

## New Kingdom specimens

We sampled three tissues—enamel, bone, and hair—from EA6738; and, significantly, we detected contrasting $^{87}$Sr/$^{86}$Sr ratios between the enamel (0.707431), a tissue that formed early in life (before 3 years of age), and bone (0.707768), a tissue that is either entirely diagenetic or reflective of the final 5–10 years of life. The former ratio lies beyond the range of Nile sediments, whereas the latter is indistinguishable from those of New Kingdom Egyptians (***Figure 4***). This result is important for two reasons: it demonstrates that EA6738 was (i) born outside of Egypt and (ii) imported into Egypt where it lived for many years. Thus, we infer that the hair of EA6738—with a mean $\delta^{18}$O value of 16.4 ± 1.2 ‰ (n = 3 replicates)—is a reflection of the water that it drank under captive conditions.

Crucially, the hair of another specimen (EA6736) is substantially more enriched in $^{18}$O, with a mean $\delta^{18}$O value of 19.2 ± 0.4 ‰ (n = 5 replicates). The magnitude of this difference is telling, for it indicates the likely retention of geoprovenance-reflecting $\delta^{18}$O values; that is, it would appear that the death and mummification of EA6736 occurred soon (days-to-months) after its arrival in Egypt. Thus, our dual-isotope ($\delta^{18}$O; $^{87}$Sr/$^{86}$Sr) approach to isomapping was based on the hair of EA6736 and enamel of EA6738, respectively.

To estimate the geoprovenance of these tissues, we derived spatially distributed estimates of $\delta^{18}$O and $^{87}$Sr/$^{86}$Sr from the tissues of modern baboons (***Appendix 1—tables 1*** and ***2***), and calculated normalized differences against each target tissue, the hair of EA6736 (***Figure 5a***) and enamel of EA6738 (***Figure 5b***). It is evident that variation in $\delta^{18}$O does little to constrain our results geographically, but it does refine our sparse sample of bone- and enamel-derived $^{87}$Sr/$^{86}$Sr ratios. Accordingly, we combined the normalized differences of both isotope ratios to visualize an area within 1 SE of both target tissues (***Figure 5c***). The area encompasses much of present-day Ethiopia, Eritrea, and Djibouti, and portions of Somalia and Yemen.

A limitation of this analysis is our use of a baboon-derived isoscape, which excludes areas now devoid of baboons, such as Nubia in northern Sudan and southern Egypt. During antiquity, it is

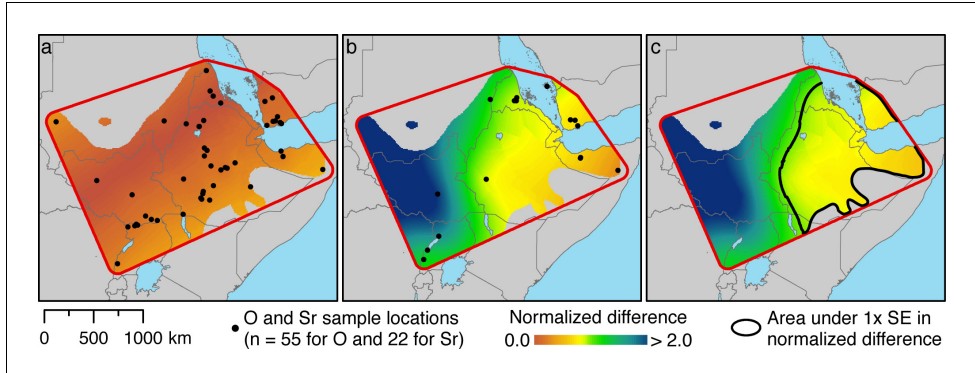

**Figure 5.** Spatial estimation of isotope ratios and their differences from target values using Empirical Bayesian Kriging. (a) Specimen locations of modern baboons (black points; source: *Appendix 1—table 1*) and the normalized difference values for δ¹⁸O against our target tissue, the hair of EA6736. (b) Specimen locations of modern baboons (black points; source: *Appendix 1—table 2*) and the normalized difference values for ⁸⁷Sr/⁸⁶Sr ratios against our target tissue, the enamel of EA6738. (c) The combined normalized difference of both isotope ratios against both target tissues. The black line bounds the area within 1 SE. Some points in panels (a) and (b) include multiple samples.

plausible that this region accommodated wild or captive populations of export-ready baboons. To explore whether EA6738 could have origins in ancient Nubia, we produced a spatial model (*Appendix 1—figure 4*) using the global bioavailable strontium isoscape of *Bataille et al., 2020*. This exercise rules out ancient Nubia as a source of EA6738, but see *Box 3* for further discussion.

## Geoprovenance and Punt

A strength of our result is that it puts EA6738 squarely within the natural distribution of *P. hamadryas* (*Figure 2*). The significance is twofold: first, it bears witness to the astounding reach of Egyptian seafaring during the 2nd millennium BC; and second, it corroborates reports of long-distance trade with Punt, a toponym of enduring fascination and debate.

Punt occupied a region south and east of Egypt, and was accessible by land or sea. For Egyptians, Punt was a source of 'marvels', particularly aromatic resins, that drove bidirectional trade for 13 centuries (ca. 2500–1170 BC; *Figure 2*). Some scholars view this commercial enterprise as the beginning of globalization (*Fattovich, 2012*), whereas others describe it as the beginning of the spice route (*Keay, 2006*), a trade network that would shape geopolitical fortunes for millennia. The global historical importance of Punt is therefore considerable, but there is a problem, which *Phillips, 1997* put succinctly: "Punt has not yet been located with certainty on any map and no archaeological remains have ever been identified even tentatively as Puntite".

Egyptologists of the 19th century were enthralled with Punt, producing at least 54 arguments for its location (*Breyer, 2016*), but it was the work of *Mariette, 1875* that ushered consensus. He put Punt on the Somali coast, an area that produces premium resin from *Boswellia frereana* (frankincense). A legacy of this era is the modern autonomous state of Puntland in northeast Somalia. *Herzog, 1968* broke with this view when he argued for a location in southern Sudan and northern Ethiopia, between Atbara and the confluence of the White and Blue Niles. Kenneth Kitchen then extended Punt eastward to the Red Sea coast of present-day Eritrea (*Kitchen, 1971*; *Kitchen, 1993*; *Kitchen, 2004*), a stretch of Africa favored by most scholars since (*O'Connor, 1982*; *Sleeswyk, 1983*; *Fattovich, 1993*; *Bradbury, 1996*; *Balanda, 2005*; *Kalb, 2009*; *Breyer, 2016*; *Bard and Fattovich, 2018*). Some authors have argued for locations as distant as Lake Albert, Uganda (*Wicker, 1998*) or Mozambique (*Lacroix, 1998*), but these claims have met with strong criticism or refutation (*Phillips, 1999*; *Bard and Fattovich, 2018*). Lastly, sound reasoning exists for the Arabian Peninsula, with Punt representing the whole eastern Red Sea coast as far as present-day Yemen (*Meeks, 2003*; *Tallet, 2013*).

Nonhuman primates are germane to this debate because Punt was a major emporium for monkeys. The pyramid causeway of Sahure (ca. 2480 BC) depicts the earliest known expedition to Punt, and monkeys are among the imported goods (*El Awady, 2009*). Literature provides another

## Box 3. Considering a Nubian origin for EA6738.

Ancient Nubia is divided into Lower Nubia (including the area between the First and Second Nile Cataracts, referred to as Wawat) and the Kingdom of Kush based at Kerma, Upper Nubia. Much of Nubia may have accommodated baboons during the African Humid Period of the early Holocene, before the shift to hyper-arid conditions around 4500 years ago. Scholars have long presumed that relics of this former distribution survived into antiquity, and that Nubians captured or raised local baboons for export. But supporting evidence is equivocal, including inscrutable rock art near the Fourth Nile Cataract (**Paner and Borcowski, 2005**) and accounts of Nubian tribute in the tomb of Rekhmire (**Figure 1d**) and Papyrus Koller (**Gardiner, 1911**), neither of which rule out Nubia as an entrepôt for trade goods sourced elsewhere. Yet, the enamel of EA6738 has an $^{87}$Sr/$^{86}$Sr ratio that lies comfortably in the range of values reported for floodplain sediments and human and animal tissues from ancient Nubia (**Buzon et al., 2007**; **Buzon and Simonetti, 2013**; **Woodward et al., 2015**), raising the possibility of a Nubian origin for EA6738. This premise is weakened, however, by Egyptian occupations of Lower Nubia during the Middle Kingdom and again during the New Kingdom, when much of Kush fell to Thutmose I during a military campaign that struck deep into Upper Nubia, reaching Kurgus (**Davies, 2005**). Had *P. hamadryas* existed in Wawat or Kush during these periods of Egyptian control, it would have been readily available for export. Yet, there are no physical specimens of *P. hamadryas* in Egypt prior to the reign of Queen Hatshepsut (**Figure 2**), the first New Kingdom monarch to resume maritime trade with Punt (**Creasman, 2014**). We are mindful of the absence-of-evidence fallacy of logic, but it is telling that imports of *P. hamadryas* from Punt are featured so prominently on the walls of Queen Hatshepsut's mortuary temple (**Figure 1c**), for it suggests a significant achievement that postdates the conquests of Wawat and Kush. On balance, there is little reason to impute a Nubian origin for EA6738.

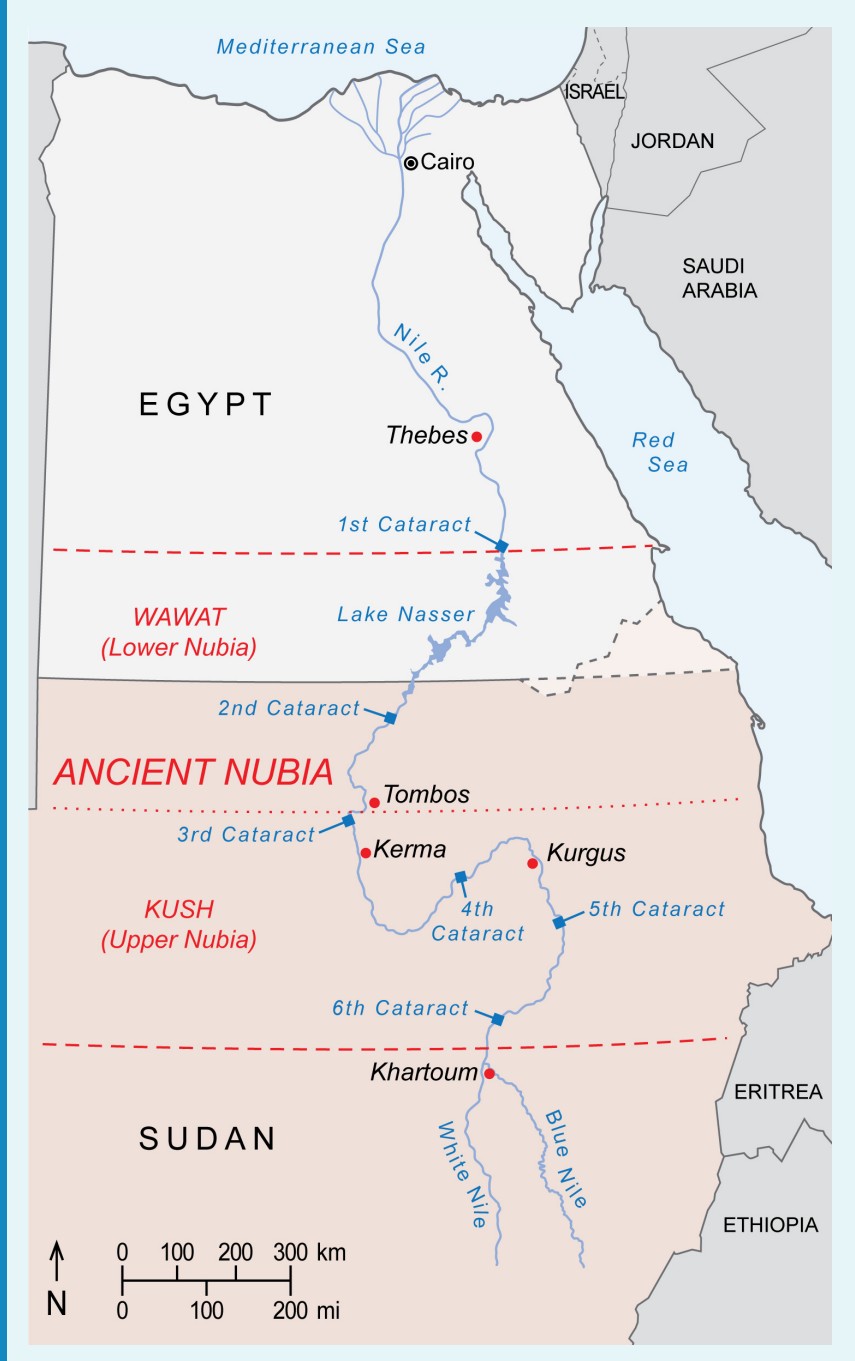

**Box 3—figure 1.** Borders of ancient Nubia (orange dashed lines) and present-day political borders.

example. In *The Tale of the Shipwrecked Sailor*, a story dated to the Middle Kingdom, an Egyptian sailor is washed ashore on a magical island in the Red Sea. There he meets a serpent identified as the 'Lord of Punt'. When the sailor is rescued, the serpent presents him with many gifts, including long-tailed monkeys and baboons (**Phillips, 1997**). Speculation that shipwrecked Egyptian sailors

were responsible for introducing *P. hamadryas* to the Arabian Peninsula is a testament to the curious distribution of this species (*Kummer, 1981*), but the idea is now refuted (*Wildman et al., 2004*; *Winney et al., 2004*; *Fernandes, 2009*; *Kopp et al., 2014*).

New Kingdom expeditions to Punt imported living specimens of *P. hamadryas*, as depicted on the reliefs of Deir el-Bahri (*Figure 1c*) and the tomb-chapels of high officials, dating from Tuthmose III to Amenhotep III. Thus, if the New Kingdom attributions of EA6736 and EA6738 are correct (and the specimens are contemporaneous with voyages to Punt) then our findings corroborate the balance of evidence that puts Punt in (i) the Horn of Africa and (ii) a broader realm, 'God's Land' (*Bradbury, 1996*), that may encompass the eastern and western coasts of the southern Red Sea (*Balanda, 2005*; *Cooper, 2011*). This possibility is attested by Egyptian texts that mention sacred baboons in another Red Sea region called Wetenet, an enigmatic toponym frequently mentioned as the origin of the solar birth and sunrise, evidently in the environs of Punt (*Cooper, 2017*). It would seem plausible or even probable that Egyptians distinguished between African and Arabian populations of *P. hamadryas*, but resolving this question is a priority for future research.

## Greco-Roman traffic in baboons

Our effort to determine the geoprovenance of five Ptolemaic baboons failed to bear fruit, but thousands of additional specimens exist in the subterranean galleries of North Saqqara and Tuna el-Gebel. It is likely that many of these animals were bred in captivity, but it is equally likely that some were imported via the Red Sea, which raises the possibility of using Sr isotope analysis to establish primatological continuity between Punt and the rise of Axum as the principal supplier of African goods to Roman Egypt (*Phillips, 1997*). Agatharchides of Cnidus (~145 BC) described the shipment of baboons from Ethiopia-Eritrea to Alexandria (*Burstein, 1989*), as well as cepi (probably patas monkeys, *Erythrocebus patas* [*Burstein, 1989*]) and sphinx monkeys (probably geladas, *Theropithecus gelada* [*Jolly and Ucko, 1969*]). Pliny the Elder named the source port—Adulis (near present-day Massawa, Eritrea)—in his *Naturalis Historia*, whereas the Egyptian port of Berenice—then a central hub in the maritime spice route, connecting East and South Asia with Egypt and the Mediterranean Basin (*Sidebotham, 2011*)—is linked to the transshipment of baboons ('cenocephali') on the *Tabula Peutingeriana* (*Geissen, 1984*).

Berenice lies ~250 km south of Mersa Gawasis, a Middle Kingdom harbour and port for seafaring to Punt (*Figure 2*). It has a rich archaeological record, including ceramics from the Gash lowlands, Eritrea, and Yemen, as well as botanical remains from the coastal plains (and immediate hinterland) of Eritrea, from Aqiq to Adulis (*Bard and Fattovich, 2018*). It is tempting, then, to suggest that Greco-Roman traffic in baboons between Adulis and Berenice merely followed in the wake of Egyptian sailors navigating between Punt and Mersa Gawasis some 2000 years earlier.

## Conclusion

Our effort to map the fabled land of Punt should be viewed as provisional. It is evident, however, that ancient Egyptians venerated *P. hamadryas* and traveled great distances to acquire living exemplars. Yet, the distribution of baboons is often overlooked when scholars discuss the location of Punt or the luxury goods that drove the evolution of international maritime commerce. Our results suggest that *P. hamadryas* was an important, contributing factor to the rise of Red Sea trade during the 2nd millennium BC.

## Materials and methods

Tissues from mummified baboons were sourced from the British Museum (*Figure 3*) and the Petrie Museum of Egyptian Archaeology, University College London (n = 5 bone fragments, each from the Baboon Catacomb of North Saqqara; [*Appendix 1—figure 3*]). We limited sampling to detached tissue fragments contained in specimen boxes, excepting the hair of EA6736, which was cut directly from the specimen (length: ~2 cm). Enamel fragments from EA6738 (*Appendix 1—figure 5*) were re-fit to the source tooth to verify association. Similarly, hair and bone samples of modern baboons were sourced from the American Museum of Natural History (New York, USA), the Field Museum (Chicago, USA), the Museo di Storia Naturale di Firenze, La Specola (Florence, Italy), the National Museum of Natural History (Washington, D.C., USA), the Natural History Museum (London, UK), and the Powell-Cotton Museum (Quex Park, UK). Additional tissues samples were supplied by colleagues

(Clifford J. Jolly, New York University; Derek E. Wildman, University of South Florida) or collected by ourselves in the field.

## Sample preparation and analysis

Oxygen isotope ratios are presented as δ values, where $\delta = 1000[(R_{sample}/R_{standard}) - 1]$ and $R = {}^{18}O/{}^{16}O$; the reference standard is Vienna standard mean oceanic water (VSMOW). Units are expressed as parts per thousand (‰). To measure the $\delta^{18}O$ of hair, two to three strands were cleaned of debris, washed three times in a 2:1 mixture of chloroform:methanol, cut into 3 mm-segments, and weighed (150 ±15 µg) into pre-combusted silver foil capsules. Next, the samples were vacuum dehydrated for a minimum of 6 days to remove oxygen under active exchange with atmospheric water vapor (*Bowen et al., 2005a*; *Bowen et al., 2009*). The dried samples were immediately combusted and analyzed with a Thermo-Chemical Elemental Analyzer (TCEA) interfaced with a Delta Plus XP isotope ratio mass spectrometer (IRMS, Thermo Finnigan, Bremen, Germany) located in the Stable Isotope Laboratory of the University of California, Santa Cruz. Analytical precision (±1 SD) based on 65 IAEA-601 (Benzoic acid) replicates was 0.1‰ and the repeated analysis of a check reference for keratin (local horse hair) was 0.3 ‰.

Samples of enamel or bone were rinsed with DI $H_2O$ to remove surface contaminants. The samples were then weighed and dissolved in concentrated $HNO_3$ and 6 M HCl in closed Teflon vials, and dried down. The samples were then re-dissolved in 8 M $HNO_3$, aliquoted, dried down, and again dissolved in 200 µl of 8 M $HNO_3$. This solution was loaded onto Sr specific resin (Eichrom) contained in Teflon micro-columns, eluted with 8 M $HNO_3$, and the Sr collected with DI $H_2O$. A drop of $HClO_4$ was then added to the Sr cut which was then dried down at ~150°C. The Sr was taken up in one drop concentrated $HNO_3$ and dried down in preparation for loading for TIMS (thermal ionization mass spectrometry). For TIMS analysis, the Sr samples were taken up in 10% phosphoric acid and dried along with a TaCl emitter solution on outgassed zone refined Re filaments. Sr isotopic measurements were accomplished with a Thermo Scientific Triton multi-collector TIMS instrument located in the Center of Isotope Geochemistry (associated with Lawrence Berkeley National Laboratory and UC Berkeley), and used in static mode (200 ratio cycles measured per analysis). Measured $^{87}Sr/^{86}Sr$ was normalized to a $^{86}Sr/^{88}Sr$ ratio of 0.1194. During the course of analysis, the NBS987 Sr isotopic standard (at least one per barrel) gave a $^{87}Sr/^{86}Sr$ of 0.710253 ± 0.000007 (±2 s, external, n = 8).

## Spatial analysis of isotope ratios

The spatial analysis of isotope ratios was conducted within a 100 km buffer region surrounding the polygonal convex hull defined by the location of each tissue sample at the time of collection in the wild. It was further constrained to the mapped range of *P. hamadryas* and *P. anubis* as shown in *Box 1*, which is based on the distributions of *Zinner et al., 2013*, *Fischer et al., 2017*, and *Singleton et al., 2017*. Within this region, Empirical Bayesian Kriging (EBK) was used to derive spatially distributed estimates of $\delta^{18}O$ for each hair sample and $^{87}Sr/^{86}Sr$ for each bone or enamel sample, using the EBK geoprocessing tool in ArcGIS 10.7 (*ESRI, 2019*). This Bayesian method of kriging does not require specification of the prior distributions of model parameters and is able to handle non-stationary data (*Krivoruchko and Gribov, 2014*). For both the $\delta^{18}O$ values and $^{87}Sr/^{86}Sr$ ratios, a K-Bessel detrended semivariogram model type was used with a local model area overlap factor of 5, a 10-km grid spacing, and a circular search neighborhood including 10 to 15 neighboring points. This process yielded estimated $\delta^{18}O$ values and $^{87}Sr/^{86}Sr$ ratios and their uncertainties for each grid cell (*Appendix 1—figures 6* and *7*). These values were then transformed into normalized differences from the two target values, the hair of EA6736 (19.2‰) and the enamel of EA6738 (0.707431), by subtracting the target values and dividing by the mean standard error of prediction from the EBK process (2.2 for $\delta^{18}O$ and 0.0011159 for $^{87}Sr/^{86}Sr$). The two normalized differences were combined using the root sum of squares (RSS) to produce the overall normalized difference metric shown in *Figure 5c*.

## Acknowledgements

We acknowledge the assistance of N Abraha, P Agnelli, D Andreasen, D Antoine, A Berhanu, AM Borgstom, D Brandon-Jones, GAO Britton, EE Butler, V Camomilla, AJ Cunningham, D Deutsch,

PF Dorman, N Fourie, S Gippoliti, L Gordon, R Johnson, CJ Jolly, HL Kafka, MI Khaled, M Harman, T Heath, LR Heaney, A Khattab, JJ and PA Lia, Y Libsekal, M Lowe, DP Lunde, S Pancaldo, JE Phillips-Conroy, S Quirke, JD Ray, JW and MD Ridges, R Sabin, WT Stanley, JH Taylor, I Tesfamariam, DE Wildman, JD Yeakel, and R Young. Archaeological site visits were approved by the Supreme Council of Antiquities (Egypt, permit no. 022569) and the Ministry of Tourism (Eritrea, permit no. DSA-M5. A2). Sampling from wild populations was approved by the Dartmouth Institutional Animal Care and Use Committee (protocol no. 10-16-13), the Uganda National Council for Science and Technology (permit no. NS-267), and the Uganda Wildlife Authority (permit no. FOD/33/02). Samples were exported under CITES permit nos. 314933/01 and 478050/01. Support for the Center of Isotope Geochemistry at LBNL is provided by the Department of Energy, Office of Basic Energy Sciences through contract DE-AC02-05CH11231 to Lawrence Berkeley National Laboratory. Additional funding was received from the Council of American Overseas Research Centers (Multi-Country Research Fellowship to NJD), Dartmouth College (the Professor Arthur M Wilson and Mary Tolford Wilson Faculty Research Fellowship award to NJD; the Women in Science Project), the David and Lucile Packard Foundation (Fellowship no. 2007–31754 to NJD), and the US-Egypt Joint Technology Fund administered by the National Science Foundation (OISE-0923655 to NJD, SI, and PLK).

## Additional information

### Funding

| Funder | Grant reference number | Author |
| --- | --- | --- |
| National Science Foundation | OISE-0923655 | Nathaniel J Dominy<br>Salima Ikram<br>Paul L Koch |
| David and Lucile Packard Foundation | 2007-31754 | Nathaniel J Dominy |
| U.S. Department of Energy | DE-AC02-05CH11231 | John N Christensen |
| Council of American Overseas Research Centers | Multi-Country Research Fellowship | Nathaniel J Dominy |
| Dartmouth College | Wilson Faculty Research Fellowship | Nathaniel J Dominy |
| Dartmouth College | Women in Science Project | Nathaniel J Dominy<br>Gillian L Moritz |

The funders had no role in study design, data collection and interpretation, or the decision to submit the work for publication.

### Author contributions

Nathaniel J Dominy, Conceptualization, Resources, Data curation, Formal analysis, Funding acquisition, Investigation, Methodology, Writing - original draft, Project administration; Salima Ikram, Conceptualization, Funding acquisition, Methodology, Writing - original draft, Writing - review and editing; Gillian L Moritz, Patrick V Wheatley, Formal analysis, Investigation, Methodology; John N Christensen, Jonathan W Chipman, Formal analysis, Investigation, Methodology, Writing - review and editing; Paul L Koch, Conceptualization, Funding acquisition, Writing - review and editing

### Author ORCIDs

Nathaniel J Dominy https://orcid.org/0000-0001-5916-418X
Paul L Koch http://orcid.org/0000-0001-5248-1529

### Decision letter and Author response

Decision letter https://doi.org/10.7554/eLife.60860.sa1
Author response https://doi.org/10.7554/eLife.60860.sa2

## Additional files

### Supplementary files

- Transparent reporting form

### Data availability

All data generated or analyzed during this study are included in the manuscript and supporting files.

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

## Appendix 1

### Monkeys as foreign tribute

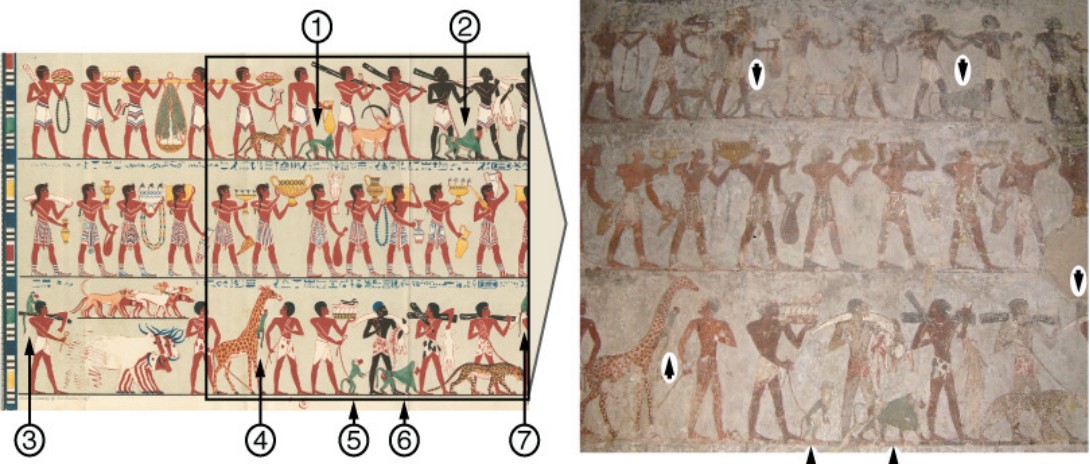

**Appendix 1—figure 1.** Monkeys were depicted as foreign tribute in the mortuary chapel of Rekhmire (TT 100), Vizier to Tuthmose III and Amenhotep II; New Kingdom, 18th Dynasty, ca. 1479–1425 BC. (*Left*) Facsimile painting by *Hoskins, 1835* (reproduced with permission from the Manuscripts, Archives and Rare Books Division, Schomburg Center for Research in Black Culture, The New York Public Library; this image is free to use without restriction). The upper, middle, and lower registers depict processions from Punt, Keftiu (Crete), and Nubia, Lower Nubia, and Khenethennefer, respectively (*Güell i Rous, 2018*). The tribute bearers from Punt and Nubia are associated with a wide range of natural resources, including seven monkeys, which Hoskins rendered as *Papio hamadryas*. (*Right*) Six of seven monkeys are detailed in a photograph by author NJD. Monkeys 2 and 6 are unambiguously *P. hamadryas*, whereas monkeys 1, 4, and 5 closely resemble vervets (*Chlorocebus aethiops*) on the basis of tail and cranial morphology.

### Further examples of canine tooth extraction

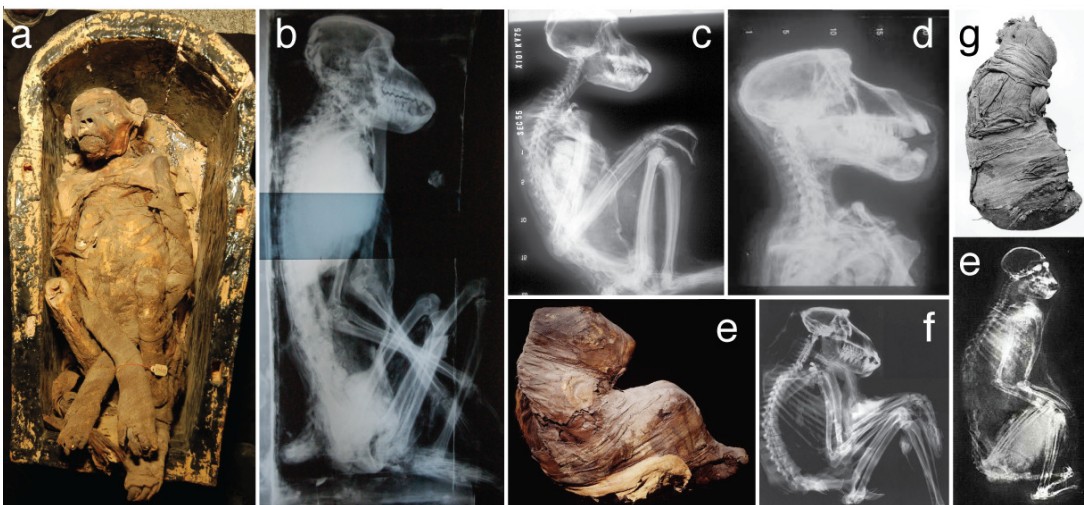

*Appendix 1—figure 2 continued on next page*

*Appendix 1—figure 2 continued*

**Appendix 1—figure 2.** Radiographs reveal the absence of canine teeth in four mummified baboons. (**a**) Specimen of *P. hamadryas* recovered from KV52, Valley of the Kings, in 1906 (accession no. MM39, Mummification Museum, Luxor). The absence of canines is evident in the corresponding radiograph of MM39 (**b**) as well as those of two males, (**c**) JE38746 and (**d**) JE38744, recovered from KV51 in 1906 and accessioned in the Museum of Egyptian Antiquities, Cairo. In general, the extraction of canine teeth is a hallmark of the New Kingdom; however, it is evident in at least one specimen (**e and f**) from the Ptolemaic period, demonstrating that the practice is imperfect evidence of a New Kingdom provenance. This specimen has no feet for unknown reasons (accession no. AMM 15). A mummified monkey (**g**) found with Maetkare (1070–945 BC), a high priestess of the 21 st Dynasty, is often described as a young specimen of *P. hamadryas* based on the radiograph (**e**) of *Harris and Weeks, 1973*; however, the animal is equipped with adult dentition and is therefore a smaller species, probably *Chlorocebus aethiops* or *Erythrocebus patas* (*Ikram and Iskander, 2002*). It is also missing its canine teeth (accession no. JE26200(e), Museum of Egyptian Antiquities, Cairo). © 2008, Rijksmuseum van Oudheden. Images in panels e and f reproduced from the Rijksmuseum van Oudheden, 2008, under the terms of the Creative Commons Attribution-ShareAlike 3.0; these images are not distributed under the terms of the CC0 1.0 license, and further reproduction should adhere to the terms of the CC-BY-SA 3.0 license.

## Late and Ptolemaic period specimens of *Papio anubis*

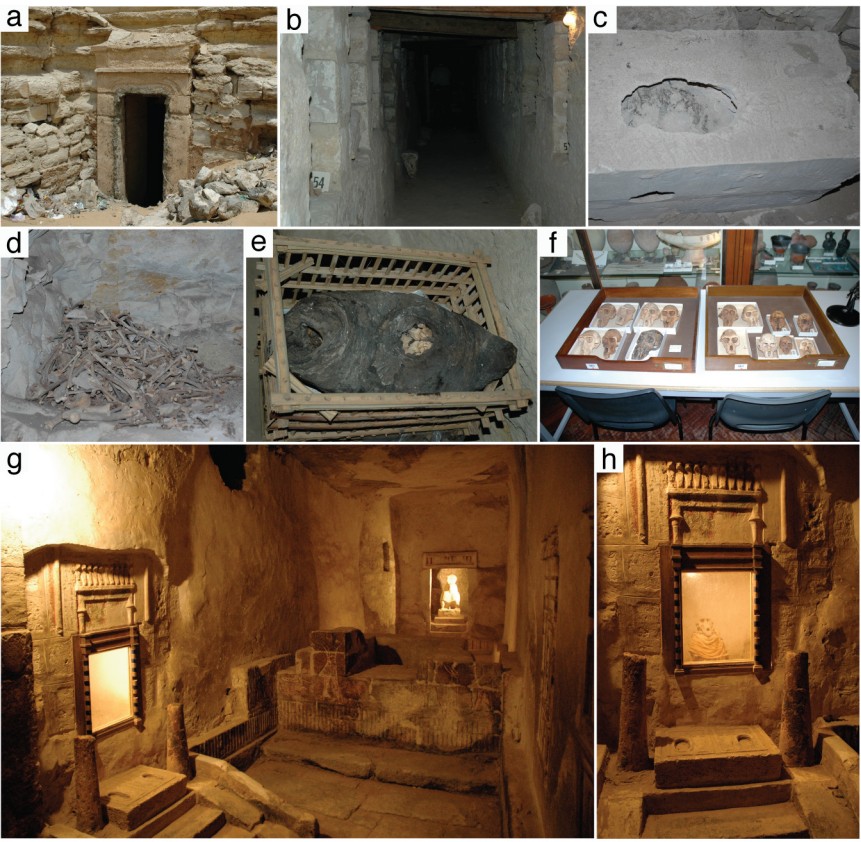

**Appendix 1—figure 3.** Sources of Late/Ptolemaic period specimens of *Papio anubis*. (**a**) Entrance to the Baboon Catacomb, North Saqqara, discovered in 1968 (*Emery, 1969*). (**b**) Niches cut into the walls of the upper gallery. (**c**) Each niche accommodated a single linen-wrapped baboon in a custom-built rectangular wooden box, infilled with gypsum plaster (*Emery, 1970*). The capacity of the catacomb was 437 burials (*Goudsmit and Brandon-Jones, 1999*). (**d**) Jettisoned osteological

*Appendix 1—figure 3 continued on next page*

*Appendix 1—figure 3 continued*

remains are present in some niches, possibly the result of Roman-era destruction. *Emery, 1969* described the wreckage as a 'frenzy of religious intolerance'. (**e**) Inventory organized by the 1996 joint expedition mounted by the Egypt Exploration Society and the University of Amsterdam. See *Davies, 2006* for detailed object descriptions. In some cases, the skeletal remains of baboons escaped destruction, existing intact and held together by thick layers of bandages. (**f**) Skulls from the Baboon Catacomb were donated to the Petrie Museum of Egyptian Archaeology, University College London, in 1969. Detached bone fragments are often present in the specimen boxes, and five such samples were analyzed here. (**g**) Chamber in the subterranean galleries of Tuna el-Gebel; the baboonification of Thoth is evident in the granite statue at the far end. (**h**) In the early Ptolemaic period, each niche was fronted by a staircase made of stone slabs, flanked by a pair of conical stone columns that acted as bases for flat bronze offering plates. Between them was a limestone offering-table for libations (*Kessler and Nur el-Din, 2005*).

Photographs by author NJD.

## Strontium isoscape of Nubia

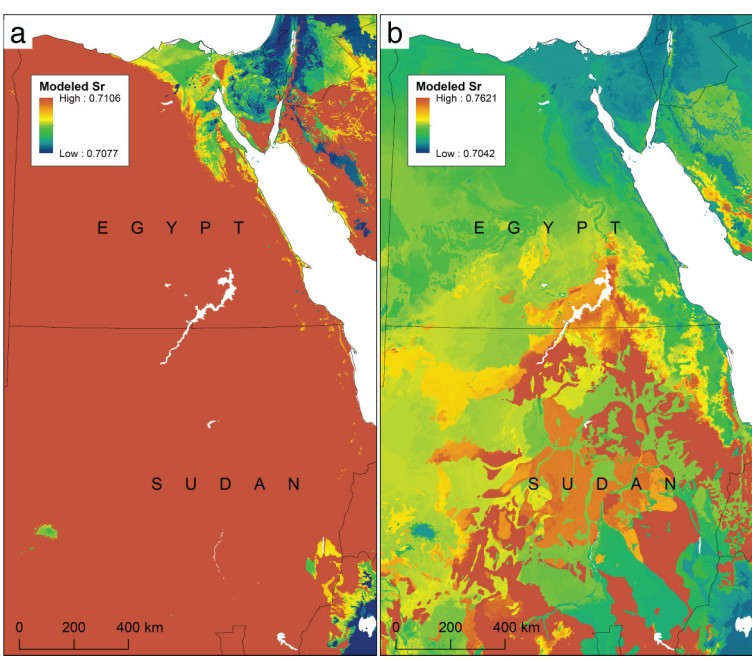

**Appendix 1—figure 4.** Recent publication of a global bioavailable strontium isoscape motivated us to consider whether the enamel of EA6748 could have an $^{87}$Sr/$^{86}$Sr ratio (0.707431) that corresponds to areas now devoid of baboons, such as Nubia in northern Sudan and southern Egypt. (**a**) Spatial model of the Sr isoscape based on the data set of *Bataille et al., 2020*, with values mapped using the same color gradient and range of Sr values in our own spatial model (*Appendix 1—figure 7*). (**b**) Same, but with a range based on the global mean ±2 SD in the data set of *Bataille et al., 2020*.

## Enamel fragments of EA6738

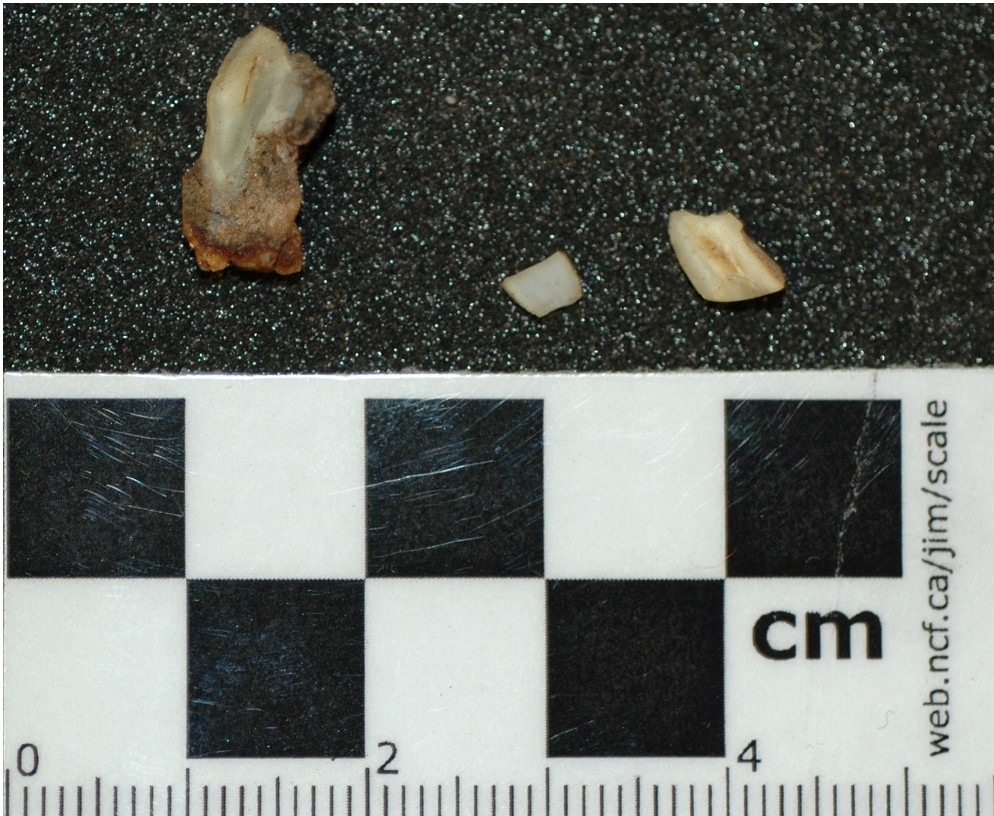

**Appendix 1—figure 5.** Enamel fragments associated with EA6738. The detached fragments were found in the specimen box and re-fit to the source tooth to verify association.

## Spatial estimations and error

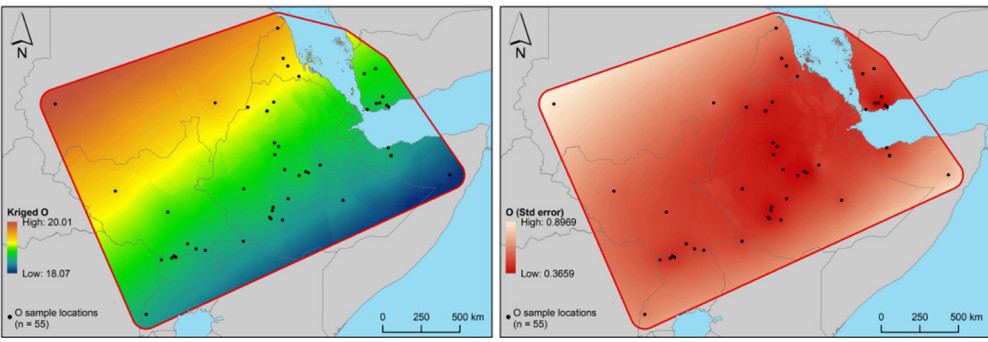

**Appendix 1—figure 6.** Spatial estimation of $\delta^{18}O$ values (*left*) and the spatially distributed estimated error of prediction (*right*) from Empirical Bayesian Kriging. Black points indicate specimen locations of modern baboons (n = 55; some points include multiple samples).

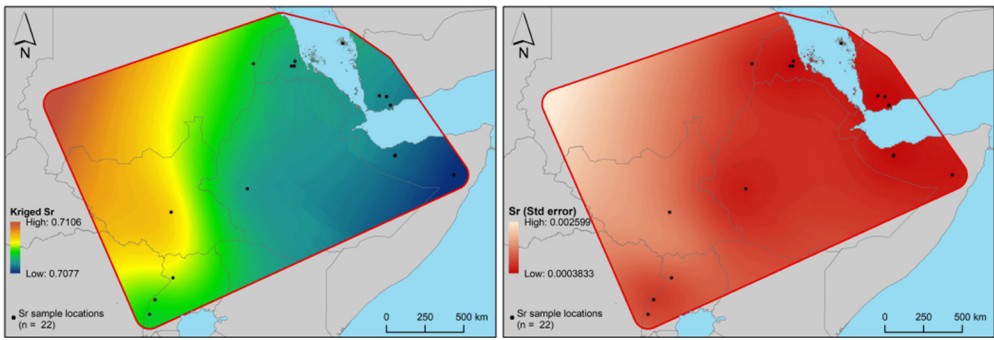

**Appendix 1—figure 7.** Spatial estimation of $^{87}Sr/^{86}Sr$ ratios (*left*) and the spatially distributed estimated error of prediction (*right*) from Empirical Bayesian Kriging. Black points indicate specimen locations of modern baboons (n = 22; some points include multiple samples).

**Appendix 1—table 1.** Sources and identifications of baboon hair samples, together with corresponding oxygen isotope values ($\delta^{18}O$; ‰, VSMOW) and geoprovenance, as described by collectors, and converted to present-day country names and geographic coordinates in decimal degrees (°, + = N, - = S for Latitude; + = E for Longitude).

| Source | Accession/ID | $\delta^{18}O$ | Source description 1 | Source description 2 | Country | Latitude | Longitude |
|---|---|---|---|---|---|---|---|
| NJD | QENJD7 | 19.8 | Queen Elizabeth National Park | | Uganda | −0.010358 | 30.0028 |
| AMNH | 184247 | 18.6 | W Nile Dist. | Koboko Co., Utukiliri | Uganda | 3.4 | 30.95 |
| BMNH | 1931.4.1.1 | 19.9 | West Madi, N.P. | Pulu | Uganda | 3.5 | 31.5833 |
| BMNH | 1937.7.24.1 | 20.1 | Laropi, Moyo | West Madi | Uganda | 3.56667 | 31.85 |
| AMNH | 184248 | 15.2 | W Nile Dist. | W. Madi, Moyo | Uganda | 3.64967 | 31.7239 |
| PCM | Uganda | 17.4 | Kedef Valley | W. of Dodinga | Sudan | 4 | 33.6667 |
| FMNH | 67015 | 16.7 | E Equatoria | Torit 50 mi SE; Ikoto | Sudan | 4.1 | 33.1 |
| FMNH | 67016 | 19.2 | E Equatoria | Torit 50 mi SE; Ikoto | Sudan | 4.1 | 33.1 |
| NMNH | 299672 | 17.6 | Al-Istiwa'Iyah Ash-Sharqiyah | Torit | Sudan | 4.40806 | 32.575 |
| BMNH | 1900.11.7.1 | 19.0 | R. Omo, L. Rudolf | | Ethiopia | 4.58333 | 36.05 |
| FMNH | 32906 | 18.0 | Sidamo | Boran Gatelo | Ethiopia | 5.91667 | 38.4667 |
| BMNH | 1967.1152 | 19.1 | R. Cullufu | between L. Abaya and L. Chamo | Ethiopia | 6 | 37.75 |
| BMNH | 1967.1151 | 19.8 | southwest corner L. Abaya | near Arba Minch | Ethiopia | 6.08333 | 37.6667 |
| AMNH | 82184 | 16.4 | Bor to Shambe | | Sudan | 6.36667 | 31.3333 |
| BMNH | 1964.2174 | 18.6 | Near NW Lake Abaya | Gamo-Gofa | Ethiopia | 6.5 | 37.8333 |
| FMNH | 27034 | 18.9 | Arusi | Abul Casim ("Abu el Kassim') | Ethiopia | 6.7 | 37.8833 |
| FMNH | 27036 | 19.5 | Bale | Webi Shebeli | Ethiopia | 7.16667 | 42.2333 |
| FMNH | 27043 | 17.2 | Shoa, Salali | Awada R; E side; nr Mt Duro | Ethiopia | 7.23278 | 38.8147 |
| BMNH | 1902.9.2.1 | 21.2 | Waw [Wau], R. Tur [Jur] | Bahr el Ghazal | Sudan | 7.66667 | 28.0667 |
| AMNH | 81061 | 16.7 | Sidamo | Agarasalam | Ethiopia | 7.83333 | 36.0667 |
| AMNH | 81062 | 17.4 | Sidamo | Agarasalam | Ethiopia | 7.83333 | 36.0667 |
| PCM | ABYS II 60 | 21.3 | Hawash | 6 hr from Oulanketi | Ethiopia | 8.67513 | 39.4938 |

*Continued on next page*

*Appendix 1—table 1 continued*

| Source | Accession/ ID | δ¹⁸O | Source description 1 | Source description 2 | Country | Latitude | Longitude |
|---|---|---|---|---|---|---|---|
| CJJ | | 20.0 | average of 20 *P. anubis* (*Moritz et al., 2012*) | along Awash River, near falls | Ethiopia | 8.85 | 40.0667 |
| CJJ | | 19.9 | average of 23 *P. hamadryas* (*Moritz et al., 2012*) | north of Lake Basaka | Ethiopia | 8.91667 | 39.9 |
| FMNH | 8171 | 15.5 | Arusi | Menegesha | Ethiopia | 9.03333 | 38.5833 |
| FMNH | 27188 | 13.3 | Shoa, Salali | Mugher R; N bank Mulu 25 mi NW | Ethiopia | 9.33333 | 40.8 |
| MZUF | 6278 | 19.7 | Garoe | Run | Somalia | 8.8 | 48.8667 |
| BMNH | 1910.10.3.1 | 23.2 | Upper Sheikh | British Somaliland | Somalia | 9.93333 | 45.2 |
| BMNH | 1902.9.9.3 | 19.1 | Ahuillet, Kotai | West Shoa | Ethiopia | 9.95 | 37.9667 |
| FMNH | 1474 | 19.4 | Togdheer | Qar Goliis ("Golis') Mts, Shiikh Pass | Somalia | 9.96667 | 45.2 |
| CMZ | E.7543B | 17.8 | Somaliland | Berbera | Somalia | 10.4333 | 45.0167 |
| FMNH | 27035 | 18.7 | Gojjam | Bichena | Ethiopia | 10.45 | 38.2 |
| FMNH | 27041 | 17.1 | Gojjam | Jigga 5 mi E | Ethiopia | 10.6667 | 37.95 |
| FMNH | 27042 | 18.6 | Gojjam | Jigga 5 mi E | Ethiopia | 10.6667 | 37.95 |
| FMNH | 27037 | 18.8 | Gojjam | Jigga 5 mi E | Ethiopia | 10.6667 | 37.95 |
| FMNH | 27192 | 17.6 | Begemdir | Gondar | Ethiopia | 12.6 | 37.4667 |
| FMNH | 27191 | 19.0 | Begemdir | Gondar 15 mi NE | Ethiopia | 12.6167 | 37.4833 |
| BMNH | 1855.12.24.19 | 13.0 | Subaihi country | 60 miles NW of Aden | Yemen | 12.75 | 43.7 |
| FMNH | 27193 | 19.3 | Begemdir | Metemma 15 mi SE Gendoa R Camp | Ethiopia | 12.8333 | 36.2833 |
| BMNH | 1973.1811 | 15.6 | ?Aden | South Yemen | Yemen | 12.9196 | 45.0203 |
| BMNH | 1920.7.30.1 | 20.2 | Jebel Marra | Darfur | Sudan | 13 | 24.3333 |
| BMNH | 1904.8.2.1 | 20.9 | Mountainous country | behind Lahej, Aden | Yemen | 13.0167 | 44.9 |
| BMNH | 1914.3.8.1 | 22.7 | Kamisa, R. Dinder | | Sudan | 13.0833 | 34.25 |
| BMNH | 1914.3.8.2 | 22.8 | Kamisa, R. Dinder | | Sudan | 13.0833 | 34.25 |
| DEW | 104 | 20.5 | Jebel Iraf | | Yemen | 13.1167 | 44.25 |
| FMNH | 27183 | 18.7 | Begemdir | Devark 30 mi NE Gonder | Ethiopia | 13.153 | 37.883 |
| DEW | 55 | 15.3 | Salah-Taiz | | Yemen | 13.15 | 44.4667 |
| BMNH | 1902.11.22.1 | 21.2 | Azraki Ravine | | Yemen | 13.5333 | 44.65 |
| BMNH | 1939.1027 | 21.9 | Senafe | Habesch | Eritrea | 14.7167 | 39.4333 |
| DEW | 30 | 17.3 | Jebel Bura'a | | Yemen | 14.9 | 43.4833 |
| DEW | 152 | 14.9 | Sana'a | | Yemen | 15.2167 | 44.1833 |
| DEW | 106 | 15.5 | Bab Al Yemen | | Yemen | 15.2167 | 44.1833 |
| PCM | ABYS I | 18.1 | Nr. Asmara | | Eritrea | 15.3333 | 38.75 |
| MZUF | 100 | 19.7 | Ausebo | presso Keren, (Bogos?) | Eritrea | 15.7833 | 38.45 |
| PCM | Sudan | 21.0 | Jebel | Kassala | Sudan | 17.5833 | 38.1 |

Source abbreviations: American American Museum of Natural History (AMNH); Clifford J. Jolly (CJJ); Derek E. Wildman (DEW); Field Museum of Natural History (FMNH); Museo di Storia Naturale di Firenze, La Specola, University of Florence (MZUF); Nathaniel J. Dominy (NJD); National Museum of Natural History (NMNH); Natural History Museum, formerly the British Museum of Natural History (BMNH); Powell-Cotton Museum (PCM); and University Museum of Zoology, Cambridge (CMZ).

**Appendix 1—table 2.** Sources and identifications of baboon enamel and bone samples, together with corresponding strontium isotope compositions ($^{87}$Sr/$^{86}$Sr) and geoprovenance, as described by collectors, and converted to present-day country names and geographic coordinates in decimal degrees (°, + = N, - = S for Latitude; + = E for Longitude).

| Source | Accession/ ID | ($^{87}$Sr/$^{86}$Sr) | Source description 1 | Source description 2 | Country | Latitude | Longitude |
|---|---|---|---|---|---|---|---|
| MZUF | 6278 | 0.707604 | Garoe | Run | Somalia | 8.80 | 48.8667 |
| BMNH | 1927.8.14.1 | 0.709239 | Lower Sheikh | British Somaliland | Somalia | 9.98333 | 45.2167 |
| BMNH | 1910.10.3.1 | 0.707509 | Upper Sheikh | British Somaliland | Somalia | 9.93333 | 45.20 |
| AMNH | 82185 | 0.712311 | Bor to Shambe | | Sudan | 6.36667 | 31.3333 |
| PCM | Sudan II | 0.708761 | Eireirib | Red Sea Province, Kassala | Sudan | 15.45 | 36.40 |
| NJD | MFNP6 | 0.711806 | Murchison Falls National Park | | Uganda | 2.27729 | 31.4617 |
| NJD | MFNP8 | 0.711651 | Murchison Falls National Park | | Uganda | 2.27729 | 31.4617 |
| NJD | MFNP3 | 0.711659 | Murchison Falls National Park | | Uganda | 2.27729 | 31.4617 |
| NJD | SEM BAB3 | 0.709508 | Semliki National Park | | Uganda | 0.908781 | 30.356 |
| NJD | SEM BAB2 | 0.708062 | Semliki National Park | | Uganda | 0.908781 | 30.356 |
| NJD | QENP5 | 0.707549 | Queen Elizabeth National Park | | Uganda | −0.01036 | 30.0028 |
| AMNH | 81062 | 0.707375 | Sidamo | Agarasalam | Ethiopia | 7.83333 | 36.0667 |
| AMNH | 81061 | 0.707406 | Sidamo | Agarasalam | Ethiopia | 7.83333 | 36.0667 |
| PCM | ABYS I | 0.707030 | near Asmara | | Eritrea | 15.3333 | 38.75 |
| NJD | Eritrea 1 | 0.707788 | near FilFil | | Eritrea | 15.6167 | 38.9667 |
| NJD | Eritrea 2 | 0.707925 | near FilFil | | Eritrea | 15.6167 | 38.9667 |
| NJD | Eritrea 3 | 0.709306 | Asmara | | Eritrea | 15.3324 | 38.9262 |
| DEW | T650 A | 0.707922 | Jebel Sabir | | Yemen | 13.5833 | 44.20 |
| DEW | T650 B | 0.707937 | Jebel Sabir | | Yemen | 13.5833 | 44.20 |
| MZUF | 11329 | 0.708758 | Isole Farasan | Isola Kebir | Saudi Arabia | 16.7065 | 41.9076 |
| BMNH | 1902.11.22.1 | 0.708158 | Azraki Ravine | | Saudi Arabia | 13.5333 | 44.65 |
| BMNH | 1904.8.2.1 | 0.708629 | mountainous country | behind Lahej, Aden | Yemen | 13.0167 | 44.90 |
| MZUF | 1669 | 0.707928 | Balad (dintorni) | | Somalia | 2.36 | 45.39 |
| MZUF | 3267 | 0.707693 | Giohar, ex Villabruzzi | | Somalia | 2.78 | 45.50 |
| MZUF | 3016 | 0.707935 | Gelib | Isola Alessandra | Somalia | 0.49 | 42.78 |
| MZUF | 2557 | 0.707665 | Afgoi | | Somalia | 2.14 | 45.12 |
| MZUF | 2460 | 0.708416 | Jesomma | Bulo Burti | Somalia | 3.85 | 45.57 |
| AMNH | 216246 | 0.723349 | Manica and Sofala | | Mozambique | −19.20 | 34.85 |
| AMNH | 216250 | 0.720642 | Inhambane | Zinave | Mozambique | −21.67 | 33.53 |
| AMNH | 216247 | 0.724701 | Manica and Sofala | | Mozambique | −19.20 | 34.85 |
| AMNH | 216249 | 0.722069 | Inhambane | Zinave | Mozambique | −21.67 | 33.53 |

Source abbreviations: American American Museum of Natural History (AMNH); Derek E. Wildman (DEW); Museo di Storia Naturale di Firenze, La Specola, University of Florence (MZUF); Nathaniel J. Dominy (NJD); Natural History Museum, formerly the British Museum of Natural History (BMNH); and Powell-Cotton Museum (PCM). Nota bene: samples from beyond the distributions of *Papio anubis* and *P. hamadryas* were excluded from further analysis.

**Appendix 1—table 3.** Strontium isotope compositions ($^{87}$Sr/$^{86}$Sr) of detached bone fragments from skulls recovered from the Baboon Catacomb, North Saqqara, and accessioned in the Petrie Museum of Egyptian Archaeology, University College London.

| Accession number | Species | $^{87}$Sr/$^{86}$Sr | ±2 s | Notes: *Groves, 1970* |
|---|---|---|---|---|
| UC30794 | *Papio anubis* | 0.707775 | 0.000006 | Adult female |
| UC30796 | *Papio anubis* | 0.707845 | 0.000006 | Adult female |
| UC30799 | *Papio anubis* | 0.707848 | 0.000006 | Juvenile II male |
| UC30802 | *Papio anubis* | 0.707848 | 0.000006 | Juvenile I, probably female |
| UC30803 | *Papio anubis* | 0.707678 | 0.000006 | Juvenile I, sex indeterminate |
| UC30804 | *Macaca sylvanus* | 0.707851 | 0.000007 | Adult male |
| UC30807 | *Chlorocebus aethiops* | 0.708209 | 0.000008 | Adult male |

