## [Decision Letter]

**Acceptance summary:**

This study presents a creative and interdisciplinary approach to a long-standing historical question: the location of Punt, an ancient kingdom with a vast trading network. The authors seamlessly integrate archaeological findings and historical texts with their analyses of oxygen and strontium isotope mapping to trace the geographical origins of mummified hamadryas baboons from ancient Egypt. Their isotopic evidence independently corroborates other historical sources pointing to the location of Punt at the horn of Africa along the Red Sea.

**Decision letter after peer review:**

Thank you for submitting your article "Mummified baboons reveal the far reach of early Egyptian mariners" for consideration by *eLife*. Your article has been reviewed by two peer reviewers, and the evaluation has been overseen by a Reviewing Editor and Christian Rutz as the Senior Editor. The following individuals involved in the review of your submission have agreed to reveal their identity: Julien Cooper (Reviewer #2); Jamie Hodgkins (Reviewer #3).

The reviewers have discussed their reviews with one another, and the Reviewing Editor has drafted this decision letter to help you prepare a revised submission.

Summary:

The authors present a creative and fascinating multidisciplinary study on the history of baboon trade and captive breeding in the ancient world. They bring a much needed scientific approach to addressing a long-standing debate about the location of Punt, an ancient kingdom with a vast trading network. Specifically, they use strontium and oxygen isotope mapping to reveal the geographical origins of mummified sacred baboons recovered from ancient New Kingdom and Ptolemaic sites. Although sample size was limited, the authors trace back the origins of mummified hamadryas baboons to the horn of Africa along the Red Sea, corroborating other lines of historical evidence that this was indeed the location of Punt.

Both reviewers and the Reviewing Editor found this paper to be unique and interesting, with very few changes needed before it can be accepted for publication in *eLife*. The reviewer comments largely center around including a more rounded and complete historical context for the location of Punt and the putative baboon distribution, as well as acknowledging the limitations of the study more explicitly.

Essential revisions:

1) The isotopic maps are limited to the modern range of hamadryas baboons. It would be interesting to see a map that includes 87Sr/86Sr values from a broader geographic region including areas north into Sudan and Egypt. While this is not the modern range for hamadryas baboons, is it not possible that their range varied over time? Similarly, wouldn't it have been useful to include areas where hamadryas baboons may have been kept captive and/or bred by humans? We appreciate that samples may be difficult to come by, but we would like to at least see the authors explicitly discuss the potential implications of this limitation on their results.

2) In relation to the previous point, one of the reviewers has seen photographs of hamadryas at Erkowit, not far from the Red Sea coast near Suakin, so some of the maps (Box 1 in particular) might show a more northerly extension of these species (or some sort of overlap with olive baboons?). The Erkowit population would be the closest *P. hamadryas* to Egypt. Would this population have had a similar isotopic signature to others? (We also assume that this population was more widespread in the ancient past). It's also possible the Erkowit baboons have been hunted to extinction in recent history.

3) Some scholars have entertained the existence of baboons in ancient Lower Nubia (especially the prehistoric) and neighbouring regions of the Sudan where they are now no longer found (see Roy, Politic of Trade in Lower Nubia, 266ff). See also the tomb of Djehutihetep at Debeira and Amunedjeh, depicting a baboon in a local Nubian landscape (Kush 8, p. 40; JEA 28, p. 50-52). So too, the present day Eastern and Western Desert populations of baboons in Sudan (Gebel Mara, Erkowit, Khor Langeb) seem to be relicts of wider distributions of baboons in wetter periods. Papyrus Koller also seems to demonstrate baboon imports from Nubia, specifically Kush (Gardiner, Late Egyptian Miscellanies, 112). There are some indications of baboons in the rock art of Gilf Kebir and also the Fourth Cataract in Nubia (Kleinitz, Rock art at the Fourth Nile Cataract: An overview).

In a more general tenor on the possibility of Sudanese populations, there is overwhelming archaeological and palaeoclimatic data for changing ecologies in Sudan, some of which was ongoing throughout the pharaonic period. Cattle, for instance, were still found in the Eastern and Western deserts of Sudan in the relevant periods, which argues for a much wetter climate and possibilities of ecologies that would have sustained baboons.

On this point, one might add that there was another important Red Sea placename in Egyptian texts (religious, medical, and expedition texts) from which baboons derived, called 'Wetenet' in Egyptian texts. The placename is consistently linked with baboons in religious texts -- but is also a real place reached on Egyptian expeditions (see Muller-Roth, Das Buch vom Tage, 155; Cooper 'Between this world and the Duat: The Land of Wetenet and Egyptian Cosmography of the Red Sea', in. C. Di Biase-Dyson and L. Donovan (eds), The Cultural Manifestations of Religious Experience: Studies in honour of Boyo G. Ockinga, 383-394; Edel, Beiträge zu den ägyptischen Sinaiinschriften). This might also have consequences for locating hamadryas/olive baboon populations, as some of these texts provide defining features of the baboons (dark skin, red-eyes). Adding all this discussion may be admittedly beyond the remit of the paper, but we think it is worth mentioning the placename at least with respect to the historical context/framework in the Introduction.

---

## [Author Response]

Essential revisions:1) The isotopic maps are limited to the modern range of hamadryas baboons. It would be interesting to see a map that includes 87Sr/86Sr values from a broader geographic region including areas north into Sudan and Egypt. While this is not the modern range for hamadryas baboons, is it not possible that their range varied over time? Similarly, wouldn't it have been useful to include areas where hamadryas baboons may have been kept captive and/or bred by humans? We appreciate that samples may be difficult to come by, but we would like to at least see the authors explicitly discuss the potential implications of this limitation on their results.

Thank you for this suggestion, which arrived on the heels of a new and highly relevant publication: Bataille et al., 2020.

Bataille et al. used a random forest regression model that integrates published strontium isotope data and predicts global variation in bioavailable strontium. They caution against extrapolation in "exceptionally geologically complex and data-poor regions", which is the certainly the case for Nubia (see Figure 2 of their paper), but we did so in response to this comment, producing a new figure (Appendix 1—figure 4) and discussionary text:

“A limitation of this analysis is our use of a baboon-derived isoscape, which excludes areas now devoid of baboons, such as Nubia in northern Sudan and southern Egypt. […] This exercise rules out ancient Nubia as a source of EA6738, but see Box 3 for further discussion.”

Box 3, in turn, has been revised significantly to address the plausibility of Nubian origins, folding in many of the comments and citations below.

2) In relation to the previous point, one of the reviewers has seen photographs of hamadryas at Erkowit, not far from the Red Sea coast near Suakin, so some of the maps (Box 1 in particular) might show a more northerly extension of these species (or some sort of overlap with olive baboons?). The Erkowit population would be the closest P. hamadryas to Egypt. Would this population have had a similar isotopic signature to others? (We also assume that this population was more widespread in the ancient past). It's also possible the Erkowit baboons have been hunted to extinction in recent history.

Thank you. We are in agreement with the plausibility of this comment, and we have corrected our maps in Figure 2 and Box 1 after verifying the distribution of *P. hamadryas* on the Sudanese coastline, per Zinner et al., 2013.

3) Some scholars have entertained the existence of baboons in ancient Lower Nubia (especially the prehistoric) and neighbouring regions of the Sudan where they are now no longer found (see Roy, Politic of Trade in Lower Nubia, 266ff). See also the tomb of Djehutihetep at Debeira and Amunedjeh, depicting a baboon in a local Nubian landscape (Kush 8, p. 40; JEA 28, p. 50-52).

Thank you for alerting us to the papers by Davies, 1942, and Säve-Söderberg, 1960. In Davies’ view, the paintings in the tomb of Amunedje are “freely imitated from the tomb of Rekhmire, though by an inferior artist”. We agree with this assessment and we give the tomb of Rekhmire (Figure 1D and Box 3) precedence for supporting Nubian ties to *P. hamadryas*, though it should be remembered that trade-goods coming from Nubia did not necessarily originate in Nubia. Nubia itself served as an entrepot for goods from other parts of Africa. The tomb of Djehutihetep at Debeira is fascinating, but Säve-Söderberg’s sketch is decidedly ambiguous. To our eye, and based on the length of the tail, the primate is a vervet / green monkey in the genus *Chlorocebus*. Indeed, Säve-Söderberg himself seems conflicted, describing the animal alternately as a “monkey” or “baboon” before stressing disparities between it and typical depictions of baboon-fruit interactions. In our view, neither paper rises to the level of corroborating a Nubian origin for *P. hamadryas* during the 18^th^ Dynasty.

So too, the present day Eastern and Western Desert populations of baboons in Sudan (Gebel Mara, Erkowit, Khor Langeb) seem to be relicts of wider distributions of baboons in wetter periods. Papyrus Koller also seems to demonstrate baboon imports from Nubia, specifically Kush (Gardiner, Late Egyptian Miscellanies, 112). There are some indications of baboons in the rock art of Gilf Kebir and also the Fourth Cataract in Nubia (Kleinitz, Rock art at the Fourth Nile Cataract: An overview).

Thank you for alerting us to the work of Gardiner, 1937, which led us to his earlier work translating the Papyrus Koller, published in 1911. The collection of papyri includes a letter from a high official named Paser to a Nubian chieftain, ordering him to make ready a long list of tribute objects, including baboons (Gardiner, 1911). This point and citation are now folded into Box 3 of our manuscript. But it should be remembered that trade-goods from Nubia did not necessarily originate there. Nubia often served as an entrepot for goods from other parts of Africa.

We consulted the 2012 paper by Kleinitz, but on the subject of baboons it cites the work of Paner and Borcowski, 2005. When asked if she has encountered images of baboons, Dr Kleinitz replied, “I have not seen any during my study of rock art at the Fourth Cataract. However, they are part of the Meroitic canon of motifs and we find quite a lot of them as graffiti on temple walls.”. Such images are too recent to support the premise of relict baboons inhabiting Kush or Wawat, and indeed, could depict animals brought in through trade. In the paper by Paner and Borcowski, 2005, an image that resembles a baboon is reproduced on page 115. The tail, however, is curved upwards rather than downwards, and the length is excessively long relative to body size. It is more likely a dog, not a baboon, as the ears also suggest a dog rather than a baboon, falling well within the canon of canine images in rock art of Sudan/Nubia and Egypt. If the original artist intended to illustrate a baboon, it is an equivocal likeness that cannot be attributed to either *Papio anubis* or *P. hamadryas*. Paner and Borcowski, 2005, also stress the challenge of dating rock art in the region. Taken together, we could not find compelling evidence from rock art to support the idea of natural populations of baboons in ancient Kush or Wawat. Still, the possibility is now expressed, and the paper by Paner and Borcowski is now cited, in Box 3 of our manuscript.

In a more general tenor on the possibility of Sudanese populations, there is overwhelming archaeological and palaeoclimatic data for changing ecologies in Sudan, some of which was ongoing throughout the pharaonic period. Cattle, for instance, were still found in the Eastern and Western deserts of Sudan in the relevant periods, which argues for a much wetter climate and possibilities of ecologies that would have sustained baboons.

We have edited Box 3 to describe the African Humid Period of the early Holocene, and the shift to hyper-arid conditions around 4500 years ago, as well as the possibility of baboons occupying northern Sudan and surviving into antiquity. A problem with expressing this view is that *Papio anubis* was the probable occupying species, not *P. hamadryas*, but such a statement puts us in the awkward position of responding to speculation with further speculation. In any case, Box 3 was written to acknowledge the possibility of baboons in ancient Nubia and simultaneously weigh the evidence for a Nubian origin of EA6738. In addition, it is clear that the climate was variable, as even in New Kingdom Egypt Farafra Oasis, in Egypt’s Western Desert, was known for supporting cattle herds. There is evidence for small areas with microclimates to support cattle, but neither pictorial or textual evidence support the existence of baboons in the area.

On this point, one might add that there was another important Red Sea placename in Egyptian texts (religious, medical, and expedition texts) from which baboons derived, called 'Wetenet' in Egyptian texts. The placename is consistently linked with baboons in religious texts -- but is also a real place reached on Egyptian expeditions (see Muller-Roth, Das Buch vom Tage, 155; Cooper 'Between this world and the Duat: The Land of Wetenet and Egyptian Cosmography of the Red Sea', in. C. Di Biase-Dyson and L. Donovan (eds), The Cultural Manifestations of Religious Experience: Studies in honour of Boyo G. Ockinga, 383-394; Edel, Beiträge zu den ägyptischen Sinaiinschriften). This might also have consequences for locating hamadryas/olive baboon populations, as some of these texts provide defining features of the baboons (dark skin, red-eyes). Adding all this discussion may be admittedly beyond the remit of the paper, but we think it is worth mentioning the placename at least with respect to the historical context/framework in the Introduction.

Thank you for alerting us to Wetenet and its association with baboons and Punt. We have happily folded this topic into our Discussion:

“Thus, if the New Kingdom attributions of EA6736 and EA6738 are correct (and the specimens are contemporaneous with voyages to Punt), then our findings corroborate the balance of evidence that puts Punt in (i) the Horn of Africa and (ii) a broader realm, "God's Land" (Bradbury, 1996), that may encompass the eastern and western coasts of the southern Red Sea (Balanda, 2005; Cooper, 2011). […] It is plausible or even probable that Egyptians distinguished between African and Arabian populations of *P. hamadryas*, but resolving this question is a priority for future research.”